# Fresh groundwater resources in a large sand replenishment

S. Huizer[1,2*], G. H. P. Oude Essink[1,2], M. F. P. Bierkens[1,2]

[1]Department of Physical Geography, Utrecht University, Utrecht, Netherlands
[2]Department of Subsurface and Groundwater Systems, Deltares, Utrecht, Netherlands

*Correspondence to*: S. Huizer (s.huizer@uu.nl / sebastian.huizer@deltares.nl)

**Abstract**. The anticipation of sea-level rise and increases in extreme weather conditions has led to the initiation of an innovative coastal management project called the Sand Engine. In this pilot project a large volume of sand (21.5 million m$^3$) – also called sand replenishment or nourishment – was placed on the Dutch coast. The intention is that the sand is redistributed by wind, current and tide; reinforcing local coastal defence structures and leading to a unique, dynamic

environment. In this study we investigated the potential effect of the long-term morphological evolution of the large sand replenishment and climate change on fresh groundwater resources. The potential effects on the local groundwater system were quantified with a calibrated three dimensional groundwater model, in which both variable-density groundwater flow and salt transport was simulated. Model simulations showed that the long-term morphological evolution of the Sand Engine results in a substantial growth of fresh groundwater resources, in all adopted climate change scenarios. Thus, the application

of local sand replenishments such as the Sand Engine could provide coastal areas the opportunity to combine coastal protection with an increase of the local fresh groundwater availability.

## 1 Introduction

Global sea-level rise poses a risk for coastal areas, especially when combined with an increase in the frequency and intensity of storm surges (Michael et al., 2013; Nicholls et al., 2010; Wong et al., 2014). Particularly small islands (Chui and Terry,

2013; Holding and Allen, 2015; Mahmoodzadeh et al., 2014), and low-lying deltas (Giosan et al., 2014; McGranahan et al., 2007; Oude Essink et al., 2010) are vulnerable to rising sea-levels. Many low-lying deltas such as the Mekong Delta (Vietnam) and the Ganges-Brahmaputra delta (Bangladesh) are already frequently subjected to extensive floods, leading to considerable economic losses, property damage, and in severe cases loss of life (Few and Matthies, 2006; de Sherbinin et al., 2011; UNDP, 2004). In addition, many ecosystems and inhabitants of deltas are threatened as a result of high subsidence

rates, over-exploitation of fresh groundwater resources, and contamination of coastal aquifers (Crain et al., 2009; de Sherbinin et al., 2011; Syvitski et al., 2009; UNDP, 2004). Sea-level rise and storm surges will enhance the pressure on these coastal regions (Kooi et al., 2000; Yang et al., 2013, 2015), and will likely exacerbate the loss of agricultural land, damage of ecosystems and the salinization of fresh groundwater resources (Hoggart et al., 2014; Nicholls, 2010; Oude Essink et al., 2010; Wong et al., 2014).

## 1.1 Coastal management

In order to protect the livelihood of densely populated coastal areas against climate-related impacts, a growing number of studies recognises the need for the adoption of coastal defence strategies (Giosan et al., 2014; Nicholls et al., 2010; Temmerman et al., 2013; Wong et al., 2014). Fortunately, the awareness of the threats posed by climate change is growing, and coastal defence in a number of countries – especially developed countries – have been intensified, specifically at vulnerable locations (Goodhew, 2014; Kabat et al., 2009; Sterr, 2008). One example is the Netherlands – a vulnerable low-lying country – where coastal defence systems have been reinforced on several occasions, in accordance with its long history of intensive coastal protection (Charlier et al., 2005). Centuries of continuing erosion, flooding and subsidence led first to the implementation of hard engineering methods (e.g. groynes and sea walls), and later soft engineering methods (e.g. sand replenishment or nourishment, van Koningsveld et al., 2008). Since 1990 the application of sand nourishments, particularly beach and shoreface nourishments, has become the dominant coastal defence strategy. Sand nourishments are applied on an annual basis – where necessary – to maintain the position of the coastline (Keijsers et al., 2015; de Ruig and Hillen, 1997).

## 1.2 Pilot project: Sand Engine

Since 2001, the position of the entire Dutch coastline is successfully maintained with 12 million m$^3$ of sand nourishments per year. However, the future annual volume of sand nourishments should increase if the coast is to rise with the sea-level (Deltacommissie, 2008). Research suggests that the annual nourishment volume should be raised to 20 million m$^3$ yr$^{-1}$ in the nearby future; in order to sustain the Dutch coastline in the long run (Giardino et al., 2011; de Ronde, 2008). The anticipation of a substantial growth in the annual nourishment volume incited discussions about the effectiveness of the current large-scale distribution of sand. These discussions led to the idea that concentrated (mega) nourishments could be more cost-effective than current practices, and may provide opportunities for natural dune growth, and recreation (van Slobbe et al., 2012).

The effectiveness, benefits and drawbacks of concentrated (mega) nourishments are currently being investigated with a pilot project named the Sand Engine (also called Sand Motor) (Mulder and Tonnon, 2011; Stive et al., 2013). In this project a mega-nourishment of 21.5 million m$^3$ was constructed at the Dutch coast in 2011: a few kilometres west from the city of The Hague (Fig. 1). The replenished sand will gradually be distributed along the coast by wind, waves and currents, thus incorporating natural forces in engineering methods (so called 'Building with Nature') (Slobbe et al., 2013; de Vriend et al., 2014). The effectiveness of the Sand Engine is investigated by extensive research and intensive monitoring: the surface elevation (including bathymetry) is measured frequently to gain detailed knowledge of the volume and direction of sediment transport at this local mega-nourishment (Ebbens and Fiselier, 2010; Tonnon et al., 2011). Recent measurements show that the outer perimeter of the 'hook-shaped' peninsula retreated, and the alongshore extent increased (de Schipper et al., 2016; Stive et al., 2013). Initially the Sand Engine extended approximately 1 km into the sea and was nearly 2 km wide at the

shoreline, while in September 2014 it extended approximately 800 m into the sea and was more than 3 km wide at the shoreline (Fig. 1).

## 1.3 Study objectives

The primary objective of this study is to quantify the potential effect of the Sand Engine on the regional groundwater system, particularly on fresh groundwater. During the life span of the Sand Engine the (direct) influence of the North Sea is diminished, because of the seaward displacement of the shoreline – and possibly growth of adjacent dunes. The extension of the beach-dune system and the reduction in seawater intrusion may lead to a growth of fresh groundwater resources. In combination with an increase of groundwater levels, the construction of the Sand Engine may also lead to a decline in the upwelling of saline groundwater and a decreased salt load in adjacent low-lying areas.

The long-term morphological evolution of the Sand Engine – powered by coastal and aeolian sediment transport – will also affect to local groundwater system with time. Erosion and deposition of sand will alter the position of the shoreline and the surface elevation with time, which simultaneously gives rise to dynamic changes in seawater intrusion and submarine groundwater discharge. The morphological evolution of the Sand Engine and the dynamic nature of this coastal system will probably lead to considerable changes in groundwater head and divide, the direction and velocity of groundwater flow, and the stored volume of fresh groundwater.

One of the innovative aspects of this study is to incorporate detailed predictions of the long-term morphological evolution of the Sand Engine in a 3D numerical model, which simulates variable-density groundwater flow. At the moment no studies have investigated the influence of local mega-nourishments on groundwater systems, and only a few groundwater modelling studies have incorporated a changing morphology in their calculations (Delsman et al., 2014). We also assess the effect of climate change (e.g. sea-level rise) on fresh groundwater resources in the study area, in combination with the morphological evolution of the Sand Engine. To our knowledge, no other studies have integrated the effect of the morphological evolution of coastal areas and climate change on fresh groundwater resources, and the number of quantitative studies that investigate the possibility to combine coastal defence with the protection of fresh groundwater resources are scarce (Oude Essink and Waterman, 2016; Oude Essink, 2001). However, studies on small islands have shown that great losses in the volume of fresh groundwater can occur as a result of decreases in groundwater recharge and sea-level rise, and especially small and thin lenses seem vulnerable to salinization (Chui and Terry, 2013; Holding and Allen, 2015; Mahmoodzadeh et al., 2014). In relation to the morphological dynamics of coastal regions, studies have shown that the erosion and accretion of sand can lead to substantial changes within the beach-foredune area (Bakker et al., 2012; Keijsers et al., 2014), and that climate change might exacerbate coastal erosion (FitzGerald et al., 2008; Zhang et al., 2004). Morphological developments in coastal areas can therefore have a substantial effect on fresh groundwater resources, and coastal management strategies that compensate, limit, or counteract coastal erosion or seawater intrusion may help to protect fresh groundwater resources.

The paper first describes the construction of the Sand Engine and the characteristics of the study area. It then reviews the methodology for the development of the regional groundwater model, and the model scenarios. Next, the model calibration and model results are described and examined as well as the impact of different climate scenarios on simulated fresh groundwater resources. Finally, the methodology and results are discussed, emphasizing on the limitations and implications for fresh groundwater resources.

## 2. Site description

### 2.1 Study area

The construction of the Sand engine was commissioned and designed by the executive branch of the Ministry of Infrastructure and the Environment (Rijkswaterstaat) and the provincial authority of South-Holland (Provincie Zuid-Holland) (Mulder and Tonnon, 2011). Large trailing suction hopper dredgers were used to extract sand from several sand pits in the North Sea and to transport this sand to the project site. The dredged material was stored in the hopper and deposited on the project site with three different techniques: by opening the bottom valves of the vessel on-site ("depositing"), by pumping a mixture of sand and water to the site through a pipeline ("pumping"), and by spraying a mixture of sand and water from the vessel's bow to the site ("rainbowing"). When the construction of the Sand Engine was completed (in July 2011), the area above MSL was 1.3 km$^2$ with a maximum surface elevation of 7 m MSL (Slobbe et al., 2013).

The Sand Engine peninsula is connected to the mainland by a sandy beach, and is bounded by a coastal dune area called Solleveld. Solleveld is relatively small dune area (circa 2 km$^2$; Fig. 2) with surface elevations ranging from 2 to 16 m MSL, and is used for the production of drinking water. From the start of the groundwater extractions in 1887 the demand and extraction of drinking water gradually increased from a maximum of 1 million m$^3$ per year before 1970, to a maximum of 7.5 million m$^3$ per year after 2008 (Draak, 2012). However, to be able to extract these increasing volumes of groundwater without salinization, the drinking water company started with the infiltration of surface water in 1970. The infiltrated volume of surface water is approximately equal to the volume of fresh groundwater that is extracted from the dunes. Currently the groundwater is extracted from the phreatic aquifer with almost 300 vertical pumping wells, which are located on the sides of twelve elongated infiltration basins [Fig. 2; Zwamborn and Peters, 2000].

Beyond the dunes the area gradually transforms into urban area and low-lying agricultural areas (polders). The low-lying polders have surface elevations of -1 to 1 m MSL, and act predominantly as a groundwater sink, while the urban areas are generally situated in higher areas with surface elevations between 1 and 3 m MSL. The dominant groundwater flow in the upper aquifers flows from the higher urban areas and coastal dunes toward the North Sea and the polders. The relatively low drainage level in the polders also leads to the attraction of deep saline groundwater, in addition to the drainage of local fresh groundwater.

## 2.2 Hydrogeology

The subsoil consists of unconsolidated sediment of predominantly fluviatile and marine origin, as illustrated by two geological profiles in Fig. 3. The upper part of the subsoil (10 to 30 m) consists of sand, clay and peat, which were deposited during the Holocene: primarily fine- to medium-grained sand in the higher situated dunes and urban areas, and primarily sand and clay in the low-lying areas. However, the lower section of the Holocene deposits (between -15 and -20 m MSL) consists in both areas mainly of clay and peat deposits. The underlying thick layers of fluviatile and marine sediment were deposited during the Pleistocene. It should be noted that the geological schematization of the aquifers and aquitards beneath -40 m MSL are based on a limited number of boreholes.

The conceptual fresh-brackish-saline groundwater distribution (blue striped lines; Fig. 3) are based on chloride measurements, performed at multi-level monitoring wells. Chloride measurements in Solleveld indicate that the boundary between fresh and brackish groundwater (1 TDS g $L^{-1}$) is situated between -20 and -40 m MSL, and the boundary between brackish and saline groundwater (10 TDS g $L^{-1}$) is situated between -40 and -60 m MSL. The depth of the fresh groundwater lens and the extent of seawater intrusion are controlled by head differences..

## 2.3 Monitoring

In order to timely observe changes in groundwater level, flow and quality an extensive monitoring network was implemented in Solleveld. After the construction of the Sand Engine this monitoring system was expanded and intensified in the western part of the dune area. The aim of the expansion of the monitoring system was to observe long-term changes in groundwater level, flow and quality and to observe hydrogeological effects caused by the Sand Engine and previous small-scale nourishments (Buma, 2013). The current monitoring system consists of more than 300 observation wells, where the groundwater level is measured with varying frequency (ranging from wells with hourly frequency to wells that are only read off every three months). The groundwater salinity is measured twice every year in at least 50 monitoring wells, with various methods: groundwater data loggers with measurement of electrical conductivity, electro-magnetic measurements, and analyses for chloride (Buma, 2013). Apart from measurements within the monitoring system, groundwater level measurements of 61 additional (onshore) monitoring wells were available in the national database of the Geological Survey of The Netherlands.

In addition to the expansion of the monitoring system, additional measures were taken to prevent salinization of fresh groundwater in the dunes. On the western base of the dunes a line of 28 interceptor (pumping) wells was installed in 2012 to maintain the groundwater level, and prevent any (negative) impact of the Sand Engine and previous nourishments on the extracted fresh groundwater in the dunes. These interceptor wells were not included in our study, because of a lack of information on the pumping rates and the expectation that the effects on the regional scale are small to negligible.

In order to gain specific information on the geohydrological dynamics within the Sand Engine, eight additional monitoring wells were installed with shallow filters (2 to 10 m below surface) and four monitoring wells with deep filters (16

to 20 m below surface). Since May 2014 the groundwater levels in the monitoring wells are continuously monitored with groundwater data loggers. The salinity of the groundwater is monitored with electro-magnetic measurement within all eight monitoring wells.

# 3 Method

## 3.1 Variable-density groundwater model

For the quantification of fresh groundwater resources in the study area we constructed a regional three-dimensional groundwater model, in which variable-density groundwater flow was simulated with the computer code SEAWAT (Langevin et al., 2008). SEAWAT has been developed by the United States Geological Survey (USGS), and numerous studies have used the code to simulate variably-density, transient groundwater flow (Heiss and Michael, 2014; Herckenrath et al., 2011; Rasmussen et al., 2012). In SEAWAT the governing flow and solute transport equations are coupled and solved with a cell-centred finite difference approximation. The model domain was discretised in 234 rows, 234 columns, and 50 layers, with a uniform horizontal cell size of 50 m and a varying layer thickness of 1 to 10 m (smallest thickness in upper layers, increasing in underlying layers; Fig. 4). The discretization and extent of the model were based on three criteria: minimise the effect on simulated groundwater heads and salinities in the study area, limit computation time, and optimise the calculation of the fresh groundwater volume. For the justification of the temporal and spatial discretisation we have performed a grid convergence test (Appendix A).

The model boundaries, as visualized in Fig. 1, were defined either perpendicular to the coastline (the SW and NE sides of the model), or parallel to the coastline (the NW and SE side of the model). Model boundaries that were defined perpendicular to the coastline, lie parallel to the dominant groundwater flow direction in the coastal area, and were therefore defined as no-flow boundaries. The other model boundaries were defined as illustrated in Fig. 4: 'specified-head' and 'head-dependent flux' boundary conditions (taking into account density differences), which represent the North Sea and local groundwater system, respectively. The 'specified-head' boundary conditions equalled the average level of the North Sea, and were applied to the seafloor. Local groundwater conditions were defined by a previous model simulation of the southwest of the Netherlands (Oude Essink et al., 2010). The base of the model was defined equal to the hydrogeological base of the model domain, which is approximately -170 m MSL and assumed to be a no-flow boundary.

The subsoil of the model was schematised to four aquifers and three aquitards (Fig. 4), based on borehole data, and the national geological databases REGIS II.1 (Vernes and van Doorn, 2005) and GeoTOP (Stafleu et al., 2013) of the Geological Survey of The Netherlands. The upper part of the phreatic aquifer (above -10 MSL) was subdivided into two hydrogeological zones with distinct hydraulic conductivities, because the geological data showed systematic differences in the sediment composition within the model domain: one zone coincides with most of the low-lying polders and contains predominantly clay, loam and fine sand deposits; the other zone contains most of the elevated areas of the model domain, where mainly fine to coarse sand was deposited during the Holocene. The aquifer parameters and layer elevations were

defined uniform for each hydrogeological unit, based on parameter estimations in the national hydrogeological database (Table 1). The molecular diffusion coefficient was set to $10^{-9}$ m$^2$ s$^{-1}$, and the longitudinal dispersivity was set to 0.2 m with a ratio of transversal to longitudinal dispersivity of 0.1. These values are similar to comparable groundwater models in the same region (Eeman et al., 2011; de Louw et al., 2011; Vandenbohede and Lebbe, 2007, 2012).

5        Other model parameters such as recharge, and surface water levels were defined by spatially distributed and time-average values of the current situation. The average monthly precipitation and reference evapotranspiration between 1981 and 2000 (Royal Netherlands Meteorological Institute, KNMI) were used to estimate the average seasonal (DJF, MAM, JJA, and SON) precipitation and evapotranspiration. Crop and interception factors were used to estimate the actual evaporation in different land use classes (e.g. forest, agriculture, urban areas) (Droogers, 2009; Hooghart and Lablans, 1988; Meinardi, 10    1994; Statistics Netherlands, 2008). Water levels, depths and widths of canals and ditches were provided by the Delfland Water Authority, and drainage levels were based on local knowledge and estimations from the Netherlands Hydrological Instrument model (de Lange et al., 2014; Massop and van Bakel, 2008). Information on the extraction of groundwater and the infiltration of surface water in the dune area Solleveld was provided by the drinking water company Dunea.

        In the model simulations we have used TDS, where TDS equals salinity [g TDS L$^{-1}$], and in the classification of the 15    groundwater salinity we have focused on three classes: fresh (0 – 1 g TDS L$^{-1}$), brackish (1 – 10 g TDS L$^{-1}$), and saline (10 – 30 g TDS L$^{-1}$). For the conversion of chloride to TDS we have used the linear relation between chloride and TDS in the North Sea; 1 g TDS L$^{-1}$ = 0.55 g Cl L$^{-1}$ (Millero, 2003). The North Sea TDS in the model domain was estimated at 28 g TDS L$^{-1}$ for all model simulations (density of 1020 kg m$^{-3}$), based on geo-electrical measurements in the North Sea near Ter Heijde between 1973 and 1997 (Rijkswaterstaat, 2012). This salinity concentration is smaller than the general North Sea 20    concentration (30-35 g TDS L$^{-1}$), because of the nearby freshwater discharge from the river Rhine. The TDS concentrations on the SE side of the model were defined by previous model calculations of the southwest of the Netherlands (Oude Essink et al., 2010). The TDS concentration of infiltration basins, canals and ditches were set to 0.2 g TDS L$^{-1}$, which is the average TDS concentration found in surface water within the study area. The spatial variation in the salinity of the groundwater recharge was estimated with semi-empirical equations, which were developed to predict the effects of sea spray deposition in 25    coastal areas (Stuyfzand, 2014). Based on meteorological measurements of the wind speed and wind direction at the measurement station in Hoek van Holland in the period 1971 – 2015, the estimated annual mean TDS concentration varied between 0.121 g $^{-1}$ at the coastline to 0.023 g $^{-1}$ at a distance of 5000 m from the high water line.

## 3.2 Calibration of pre-development conditions

The main purpose of the model calibration was to generate a valid representation of the pre-development conditions of the 30    Sand Engine (prior to March 2010). In order to exclude anomalous effects of recent sand nourishments on groundwater heads and concentrations, only observations prior to 2010 were included in the model calibration. We considered three calibration criteria: the error between simulated and observed groundwater head and TDS concentration should be similar or smaller than the observed variations in groundwater level (the average standard deviation of observations is 0.4 m) and

concentration (the average standard deviation of observations is 0.7 g TDS $L^{-1}$), the error should be randomly distributed in space, and the simulated distribution of the TDS concentration should correspond with literature (Stuyfzand, 1993).

The calibration comprised sensitivity analyses, (restricted) manual model parameter calibration, and comparisons of simulated groundwater heads and TDS concentrations with averaged observations of recent years. Historical processes that

promote or diminish seawater intrusion were included in the calibration, because a salinity distribution often takes decades to hundreds of years to reach an equilibrium (Delsman et al., 2014; Webb and Howard, 2011). Examples of historical processes that have substantially influenced the groundwater salinity in the Dutch coastal area are coastal erosion, sea-level rise, and groundwater extractions (Post et al., 2003). These processes were therefore included in the model simulations to attain a better match between simulations and observations, and the method of incorporation of the processes is briefly described in

Table 2 and visualised and in Fig. 5. Other historical changes in for example groundwater recharge and subsidence were not included, because measurements and historical data indicate that these processes probably have a negligible impact on the current head and concentration distribution in the study area (CBS et al., 2012; Hoofs and van der Pijl, 2002). The simulation was restricted to the period 1500 – 2010, because the focus of this study lies on the present conditions, and we assume that the most substantial effects on the present salt distribution will probably occur in this period. Before the simulation of the

period 1500 – 2010, a transient simulation of the approximate conditions in AD 1500 was executed until the model converged to a dynamically stable state in terms of both groundwater heads and salinity.

In order to attain an optimal calibration result with a limited number of model simulations, model parameters were manually adjusted with small increments from an initial best guess. The adjustments were performed on a selection of the model parameters: (horizontal and vertical) hydraulic conductivity, drainage resistance, stream bed resistance of canals and

ditches, and (longitudinal and transverse) dispersivity. Other parameters such as groundwater recharge and surface water levels were based on measurements, maps or expert knowledge, and were excluded from the calibration. The optimised model parameters that were implemented for the model scenarios are described in Sect. 3.1.

### 3.3 Morphology and climate scenarios

The effect of the Sand Engine on fresh groundwater resources will primarily depend on the morphological evolution of the

coastal area. To assess the potential effect of the mega-nourishment on coastal groundwater, we performed model simulations containing projections of the morphological change of the Sand Engine during the period 2011 – 2050 (Table 2). The morphological development of the Sand Engine in this period was simulated with the hydrodynamics and morphodynamics model code Delft3D (Lesser et al., 2004). This numerical morphodynamic model was calibrated for the period 2005 – 2010 and validated for the period 1990 – 2005, prior to the construction of the Sand Engine (Fiselier, 2010;

Tonnon et al., 2009). Based on representative tidal boundaries and wave conditions of the current situation, the morphodynamic model was used to simulate the change in bathymetry from 2011 to 2050 (Fig. 6). These simulated changes in the bathymetry were incorporated in the groundwater model by sequential grid regenerations of the model grid, for every three months (viz. season) in the simulation period. Subsequent changes in the area of inundation, groundwater recharge and

thickness of the phreatic aquifer were also adapted in the associated model input files. Estimations of the change in the mean water level in the lagoon (at the northern side of the Sand Engine), which result from the morphological changes, were also included in the model scenarios (de Vries et al., 2015). The expected maximum mean water level in the lagoon equalled 0.9 m MSL in the simulation period..

5    In addition to the morphological development of the Sand Engine, climate change may also have an impact on coastal groundwater. For the assessment of the potential impact of climate change on fresh groundwater resources, we have used the KNMI'14 climate change scenarios $G_L$, $G_H$, $W_L$ and $W_H$ (KNMI, 2014). These scenarios contain climate projections for the Netherlands for the years 2030, 2050 and 2085, based on global climate models as described in the 5th IPCC Assessment report (IPCC, 2013). The climate projections of sea-level rise, precipitation and potential evapotranspiration for 10    2030 and 2050 in these scenarios were used to assess the effect of climate change as summarised in Table 3.

All climate change scenarios were simulated for a reference case without the Sand Engine, and the current situation with the Sand Engine including the projected morphological evolution. The reference case serves primarily as a comparison to the simulations with the morphological evolution of the Sand Engine. The dissimilarity between both situations represents the total impact of the construction of the Sand Engine on local fresh groundwater resources. In turn, the climate change 15    scenarios show the response and sensitivity of local fresh groundwater resources to alterations in sea-level rise, precipitation and evapotranspiration.

## 4 Results

### 4.1 Model calibration

For the calibration of the variable-density groundwater model, we compared the simulated pre-development groundwater 20    head and TDS concentration with recent observations of groundwater heads and chloride concentrations (Fig. 7). The calibration was performed with averaged values of recent observations, and therefore transient model was strictly speaking not calibrated. We think this is acceptable, because of: the long-term scope of this study, the conservative value of 0.15 that was used for the specific yield in the simulations, and the deficiency in long-term time-series of head and especially salinity. The absolute mean error between observed and simulated heads was 0.27 m (RMSE of 0.33 m), and between observed and 25    simulated TDS concentrations was 1.17 g TDS $L^{-1}$ (RMSE of 2.75 g TDS $L^{-1}$). The largest deviations in head occur at observation points that are situated near infiltration basins or pumping wells, whereas the deviations in TDS concentration appear to be well-distributed. These deviations are probably primarily caused by heterogeneity in the phreatic aquifer and spatial variations in the extraction rates of pumping wells.

The groundwater heads of 137 observations points and the chloride concentrations of 55 observations points were 30    used to quantify the error and calibrate the model. Despite this relatively large number of observations points, it is important to note that all observations of chloride and 72% of observations of groundwater heads - that were used in the model calibration - originate from the monitoring system in Solleveld,. The simulation of the groundwater head and especially the

TDS concentration are therefore most reliable in Solleveld and the immediate surrounding system. The phreatic groundwater level and depth of the fresh-brackish interface of the calibrated model are shown in Fig. 8. Phreatic groundwater flows from the coastal dunes toward the sea, pumping wells, and low-lying drained polders. The aquitard beneath the phreatic aquifer (between -16 and -20 m MSL) limits the interaction with the underlying confined aquifer, leading to a substantial head difference across the aquitard (ranging between 0.3 to 1.4 m in multilevel monitoring wells). The fresh groundwater lens below the coastal dunes extents to approximately -30 and -40 m MSL and the interface between brackish and saline groundwater lies between -40 and -50 m MSL, corresponding with the observed depth of the interfaces (Fig. 11). Drainage in low-lying polders leads to the seepage of brackish or saline groundwater, which results in a reduction of the fresh groundwater lens thickness (Fig. 8).

In order to assess the performance of the calibrated groundwater model, we have compared simulated groundwater heads and TDS concentrations with recent observations at 8 monitoring locations on the Sand Engine (Fig. 9). The absolute mean error between observed and simulated groundwater heads was 0.36 m, and between observed and simulated TDS concentrations was 6.5 g TDS $L^{-1}$. The model appears to underestimate the hydraulic gradient – in particular in the higher regions of the Sand Engine – and groundwater salinities with a concentration higher than 15 g TDS $L^{-1}$ (between 6 and 20 m below surface). Probable causes of these discrepancies lie in the initial groundwater level and salinity (strongly influenced by the construction), the underestimation of the vertical anisotropy as a result of small mud drapes in the Sand Engine and varying weather conditions (e.g. recharge, overwash). In addition, the measured TDS concentrations are single point measurements that may not represent the average TDS concentration in the Sand Engine.

### 4.2 Fresh groundwater resources

The effect of the construction and long-term morphological evolution of the Sand Engine on the volume of fresh groundwater is initially small and similar to the situation without the Sand Engine (Fig. 10). In all model scenarios the volume of fresh groundwater slightly declines in the first years, because of the small size of the freshwater lens in the Sand Engine with respect to the cell resolution and the instability of the initial conditions. However, the gradual growth of the freshwater lens in the Sand Engine and adjacent areas eventually leads to an increase of the volume of fresh groundwater in the model domain of 0.3 to 0.5 million $m^3$ per year. This increase of the volume of fresh groundwater manifests itself mainly as an outward extension of the fresh groundwater lens in the phreatic aquifer. Underlying aquifers and aquitards may even become more saline, primarily as a result of transient boundary conditions (i.e. historical coastal erosion and on-going sea level rise) leading to continuing historical seawater intrusion. In addition, rising groundwater levels in and around Sand Engine can lead to increases in the infiltration of saline groundwater through the thin aquitard (Fig. 11 and Fig. 12).

The sea-level rise (in total) of 0.15 m in climate scenarios $G_L$ and $G_H$ and 0.25 m in climate scenarios $W_L$ and $W_H$ lead to a decline in the volume of fresh groundwater, because of the increase of seawater intrusion and inundation of the coastal area. However, the effect of sea-level rise is relatively small in respect to the total increase of fresh groundwater (Fig. 10). The long-term predictions in precipitation and evapotranspiration within the four climate scenarios (Table 3) have a

limited effect on the total volume of fresh groundwater. The climate scenarios with a strong response ($G_H$ and $W_H$) lead to a smaller volume of fresh groundwater, when compared with the climate scenarios with a weak response ($G_L$ and $W_L$). This is primarily a result of the difference in the net groundwater recharge in the climate scenarios, and the overall (yearly) volume of groundwater recharge is larger in the milder climate scenarios ($G_L$ and $W_L$). The larger increase in precipitation in winter seasons of climate scenario $G_H$ and $W_H$, coincides with a stronger increase in evaporation and a smaller increase in precipitation in the summer seasons (Table 3). However, the contrast between these climate scenarios only becomes apparent after 2030, because the precipitation and evapotranspiration patterns are equal until 2030 and diverge after 2030.

In addition to the change in fresh groundwater resources in the beach-dune system, the simulations with the long-term morphological evolution of the Sand Engine show small to negligible increases (smaller than 1 m in 2050) in the freshwater lens thickness in low-lying polders. However, changes in the total salt load in drains, canals and ditches are small in the situation with and without the Sand Engine. As a result the construction and morphological evolution of the Sand Engine may lead to small decrease of seawater intrusion, but this effect will probably be small to negligible and limited to small low-lying polders in a short distance from the Sand Engine (Fig. 13).

## 5 Discussion

The model simulations show that the construction of the Sand Engine may result in the growth of the volume of fresh groundwater by several million $m^3$. Despite the gradual erosion of the nourished sand – leading to a slow return to the previous state – the volume of fresh groundwater may continue to rise for decades after the construction of the mega-nourishment. However, tidal fluctuations and in particular storm surges will lead to land-surface inundations and consequently to a salinization of fresh groundwater. In addition, the increase in the volume of fresh groundwater is dependent on the rate of sea-level rise and the extent to which precipitation and evapotranspiration patterns will diverge from present conditions. This steady increase of the volume of fresh groundwater is in contrast with the reference case (without the construction of the Sand Engine) where historical and future sea-level rise lead to a decrease of the volume of fresh groundwater. Our results also suggest that the construction of the Sand Engine may abate the salinization of neighbouring polders, by reducing upward seepage of saline groundwater. Even though the reduction of the salinization is probably slight and limited to a small area, it might constitute an important mitigation in other applications of mega-nourishments.

Comparisons of observed and simulated groundwater heads and salinities show a good correspondence before and after the construction of the Sand Engine, despite large variations between observed and simulated groundwater salinities at individual locations. To some extent these discrepancies can be accounted for by the relatively sharp transition between fresh and salt groundwater, through which small variations in depth can result in large differences in groundwater salinities. Other factors that were not included in the simulations and that probably led to discrepancies in observed and simulated groundwater heads and salinities are: historical events (e.g. changes in groundwater level and salinity during the construction of the Sand Engine), large inundations of the Sand Engine due to storm-surges (e.g. two major storms in 2011 – 2016

inundated approximately 56% of the Sand Engine), variations in extraction rates of pumping wells, fluctuations in sea salinity and unaccounted vertical layering of the Sand Engine deposits. These factors were not included in the model simulations because of the absence or shortage of data, and the long-term scope of this study. However, the overall similarity between observations and simulations, in combination with the absence of systematic errors in the model calibration,

confirms the reliability of the model. Most of the observations – in particular groundwater salinity - emanate from the monitoring system in the adjacent dune area Solleveld and to a lesser extent the Sand Engine. The simulated groundwater heads and salinities are therefore most reliable in our area of interest, and the reliability is less in other areas in the model domain. However, the most substantial changes in groundwater salinity will take place in the area close to the Sand Engine, and variations in groundwater head and salinity in other areas will probably have a small to negligible impact on the

potential effects of the Sand Engine.

Considering the scale and nature of our research objective, we neglected small and local variations in hydraulic parameters (e.g. hydraulic conductivity, layer thickness, porosity, and storage coefficient) in the model simulations. Supported by geological data and models, each aquifer and aquitard was defined homogenous and anisotropic, with the exception of the phreatic aquifer. This reduction of the model complexity enhances the ability to differentiate and to

understand the simulated processes, and leads to a smaller computation time of the model. However, small or local variations in groundwater head or salinity that are caused by heterogeneity will not be accurately reproduced in the model simulations.

One of the largest uncertainties in the study is the long-term morphological evolution of the Sand Engine, despite extensive calibration and validation and the large number of processes that are included in Delft3D (e.g. wind shear, wave forces, tidal forces, density-driven flows). The highly dynamic nature of the coastal zone, the absence of aeolian transport in

the Delft3D simulations, and the lack of understanding of some processes, can lead to incremental differences with reality. . Even though measurements of the last four years show a reasonable fit with the projections of the sediment volume changes and erosion patterns (de Schipper et al., 2014), future morphological change can turn out to be significantly different from the morphological model. For example, the growth of dune grasses and the exposure of shell deposits may prove to reduce erosion and decelerate the morphological evolution of the Sand Engine, or an accumulation of sand in the lagoon might lead

to earlier silt up, and therefore a reduction of seawater intrusion in comparison with the projections. The implementation of one simulation of morphological change in the model calculations is therefore a significant limitation in the estimation of the potential fresh groundwater resources. For a more extensive analysis of the uncertainties in the prediction of the effects on fresh groundwater resources, it is recommended to simulate more morphological scenarios in future studies.

In addition to the long-term morphological evolution of the Sand Engine, large uncertainties also exist in the

climatological predictions of sea-level rise, precipitation and evaporation in future decades. Predictions of sea-level rise for the North Sea in 2050 range between 15 to 40 cm above MSL, and model simulations have shown that substantial changes in the growth or volume of fresh groundwater resources can occur within this range. Changes in sea-level rise and the intensity or frequency of storm surges will not only significantly influence fresh groundwater, but will also contribute to coastal erosion and alter the morphological development of the Sand Engine.

**6 Conclusions**

Local mega-nourishments such as the Sand Engine might become an effective solution for the threats that many low-lying coastal regions face, and with this study we have shown that fresh groundwater resources can substantially grow within the lifespan of the nourishment. The results in this study show that for the Sand Engine, the construction of a mega-nourishment

5   can lead to increase of fresh groundwater of approximately 0.3 to 0.5 million m$^3$ per year. However, the increase in fresh groundwater resources in a mega-nourishment is highly dependent on the shape and location of the mega-nourishment, the precipitation surplus, the frequency and intensity of storm surges, and local hydrogeological conditions. Therefore dependent on the design and location of the mega-nourishment this may provide an opportunity to combine coastal protection with the protection of fresh groundwater resources. This study also demonstrated that, with relatively simple modifications, a

10  changing morphology can easily be modelled with a variable-density groundwater model such as SEAWAT.

## Appendix A: Grid convergence test

In order to justify that the chosen spatial discretisation was adequate for reliable numerical quantifications of the potential effect of the Sand Engine on fresh groundwater resources, we have executed a grid convergence test for the period 2011 to 2050 (with and without Sand Engine). The numerical simulations that are described in this paper were performed with a horizontal grid size of 50 m, and 50 layers with a variable thickness from 1 (upper layers) to 10 m (lower layers). This spatial discretisation was tested with three additional simulations with higher and lower spatial resolutions (Table A1): one with a finer horizontal grid size of 25 m (S1), one with a coarser horizontal grid size of 100 m (S2), and one with an equal horizontal grid size of 50 m and an increased vertical resolution of the upper layers (S3). In the upper part of the model, up to a depth of -50 m MSL, the layer thicknesses were lowered with 50% (30 layers were added, up to a total of 80 layers).

Stability constraints and accuracy requirements were used for the temporal discretisation of the simulations, and therefore the convergence of the solutions with regard to the temporal discretisation was not tested. However, we have performed an additional test with respect to the coupling of the flow and solute-transport equations. The simulations were conducted with the "explicit coupling" approach. In order to test this coupling approach, an additional simulation was conducted with the "implicit coupling" approach (density criterion of 0.2 kg m$^{-3}$). For more information on these coupling approaches, we refer to the reports that describe the model code SEAWAT (Guo and Langevin, 2002; Langevin et al., 2003).

All the additional numerical simulations include no climate change scenario, and were compared to the current numerical simulations that contained a horizontal resolution of 50 m and 50 layers. The initial conditions of all additional numerical simulations were equal to the calibrated pre-development groundwater heads and TDS concentrations.

The comparison of the numerical simulations with varying spatial resolutions (Fig A1) shows a similar increase of the volume of fresh groundwater during the simulation period of 2011 to 2050. In the situation with the Sand Engine (Fig. A1b), a coarser spatial resolution lowered the projected volume of fresh groundwater (-10% in 2050), and a finer horizontal and vertical spatial resolution raised the projected volume of groundwater (respectively + 4% and +20% in 2050). However, when taking into account the deviations in the volume of fresh groundwater in the reference case (Fig. A1a), the total change in the volume of fresh groundwater becomes smaller; respectively -2%, +0% and +14% in 2050. The additional simulation with the "implicit approach" to coupling shows a small to negligible difference (smaller than 2% during the entire simulation period) with the simulations with the "explicit approach" to coupling of flow and transport equations (Fig. A2).

In order to provide a uniform measure and error analysis of the spatial grid convergence, the Grid Convergence Index (GCI) was applied to the simulated increase of the volume of fresh groundwater in the model domain in 2050. The GCI indicates to what extent the simulated increase in the volume of fresh groundwater differs from the asymptotic volume, which would be reached with further spatial refinements of the model grid. For more information on the GCI we refer to Roache (1994). The GCI of the additional model simulations S1 to S3 (Table A1) was calculated with a safety factor of 1.25. The ratio in the GCI of simulation S1 and S2 showed that the simulation was within the asymptotic range of convergence

(ratio approaches 1), which suggests that the grid was sufficiently refined. The GCI of model simulations S1 to S3 confirms that a horizontal refinement leads to small reductions in the numerical error in the volume of fresh groundwater, and that an increase in the vertical resolution could lead to a larger reduction of the numerical error. However, the refined model simulations (A1 and A3) suggest that the increase in the volume of fresh groundwater would increase with a higher spatial resolution. Therefore we think that the chosen spatial discretisation was adequate for reliable numerical estimations of the effect of the Sand Engine on the volume of fresh groundwater.

**Acknowledgements**

We thank Arjen Luijendijk and Pieter Koen Tonnon for providing Delft3D data, and performing additional Delft3D simulations. This research is supported by the Dutch Technology Foundation STW, which is part of the Netherlands Organisation for Scientific Research (NWO), and which is partly funded by the Ministry of Economic Affairs. This work was carried out within the Nature-driven nourishment of coastal systems (NatureCoast) program.

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

**Table 1.** Hydrogeologic parameters used in the simulations, where the upper part of the phreatic aquifer (1a: above -10 m MSL) was subdivided in two hydrogeological zones (1: fine to coarse sand; 2: clay, loam and fine sand)

| Layer | $K_H$ [m d$^{-1}$] | $K_V$ [m d$^{-1}$] | $\eta_e$ [-] |
|---|---|---|---|
| Phreatic aquifer 1a | 10 / 1 | 1 / 0.1 | 0.3 |
| Phreatic aquifer 1b | 1 | 0.1 | 0.3 |
| Aquitard 1 | 0.01 | 0.001 | 0.1 |
| Aquifer 2 | 30 | 10 | 0.3 |
| Aquitard 2 | 2 | 0.2 | 0.1 |
| Aquifer 3 | 5 | 2 | 0.3 |
| Aquitard 3 | 1 | 0.1 | 0.1 |
| Aquifer 4 | 15 | 3 | 0.3 |
| Aquitard 4 | 10 | 0.03 | 0.1 |

**Table 2.** Simulation of processes and method of incorporation in groundwater model

| Process | Source / Simulation | Method of incorporation |
|---|---|---|
| **Historical coastal erosion** (1500 – 2010) | Literature and paleogeographic maps of AD 1500 and AD 1850 (Beets and van der Spek, 2000; Beets et al., 1992; Vos and de Vries, 2013) | Delineation of the shoreline was incorporated in three phases: 1500, 1500-1740, 1740-2010. Phasing based on literature (Fig. 5) |
| **Historical sea-level rise** (1500 – 2010) | Literature containing time-series and predictions of historical sea-level rise (Jensen et al., 1993; Wahl et al., 2013) | The period 1500 -2010 was divided in 8 stages, enforcing the average sea-level for each stage (stages are indicated with vertical lines in Fig. 5). |
| **Groundwater extraction** (1890 – 2010) | Literature on historical groundwater extraction in Solleveld (Draak, 2012) and time-series of groundwater extraction and infiltration volumes. | The period 1890 -2010 was divided in 6 stages, enforcing the average extraction for each stage (see blue bars in Fig. 5). |
| **Morphological evolution Sand Engine** (2011 – 2050) | Simulated with Delft3D (Lesser et al., 2004), with computations of the hydrodynamics, waves, sediment transport and bed change (Mulder and Tonnon, 2011; Tonnon et al., 2009). | For every three month period in 2011 – 2050 the simulated bathymetry was enforced to the groundwater model; by changing the topography, area of inundation, and recharge. |
| **Sea-level rise** (2011 – 2050) | Climate projections of sea-level rise in 2030 and 2050 (KNMI, 2014). | The projected sea-level rise in 2030 and 2050 were linearly interpolated. The average sea-level was implemented for every three month period. |

**Table 3.** Model climate change scenarios for the period 2011 – 2050, with two rates of SLR, starting at 0.05 m MSL and seasonal variation in groundwater recharge (DJF, MAM, JJA, SON)

| Climate Scenario | Sea-Level Rise | Precipitation 2050 (given per season) | Potential evaporation 2050 (given per season) |
|---|---|---|---|
| No climate change (NoCC) | No sea-level rise [ 0.05 m MSL ] | Equal to present: Period 1981 - 2010 | Equal to present: Period 1981 - 2010 |
| $G_L$ | + 3.75 mm yr$^{-1}$ [ 2050: 0.20 m MSL ] | +3%, +4.5%, +1.2%, +7% | +2.9%, +1.3%, +3.9%, +2.7% |
| $G_H$ | + 3.75 mm yr$^{-1}$ [ 2050: 0.20 m MSL ] | +8%, +2.3%, -8%, +8% | +2.4%, +2%, +7.5%, +2.8% |
| $W_L$ | + 6.25 mm yr$^{-1}$ [ 2050: 0.30 m MSL ] | +8%, +11%, +1.4%, +3% | +3.2%, +1.7%, +4.4%, +5.8% |
| $W_H$ | + 6.25 mm yr$^{-1}$ [ 2050: 0.30 m MSL ] | +17%, +9%, -13%, +7.5% | +2.7%, +2.9%, +10.6%, +4.5% |

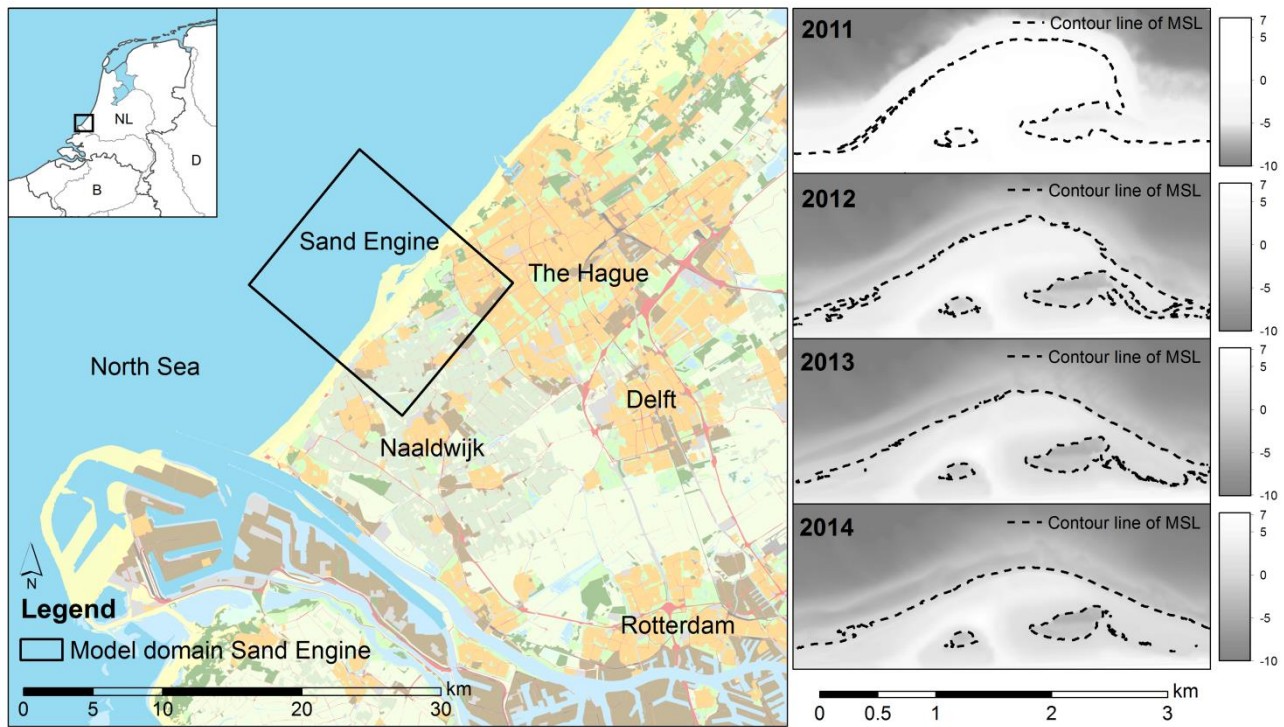

**Fig. 1.** Situation of the Sand Engine and morphological development in 2011 - 2014

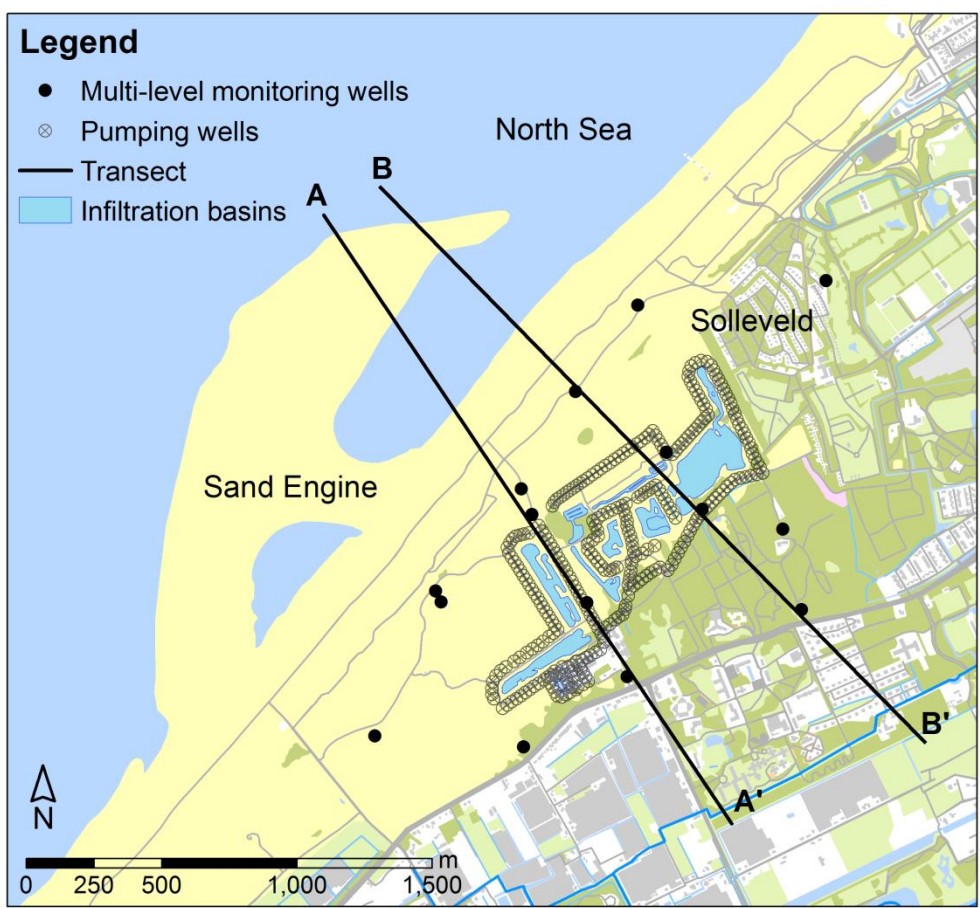

**Fig. 2.** Map of the dune area Solleveld with multi-level monitoring wells and pumping wells, including transects A and B (geological profiles in **Fig. 3**)

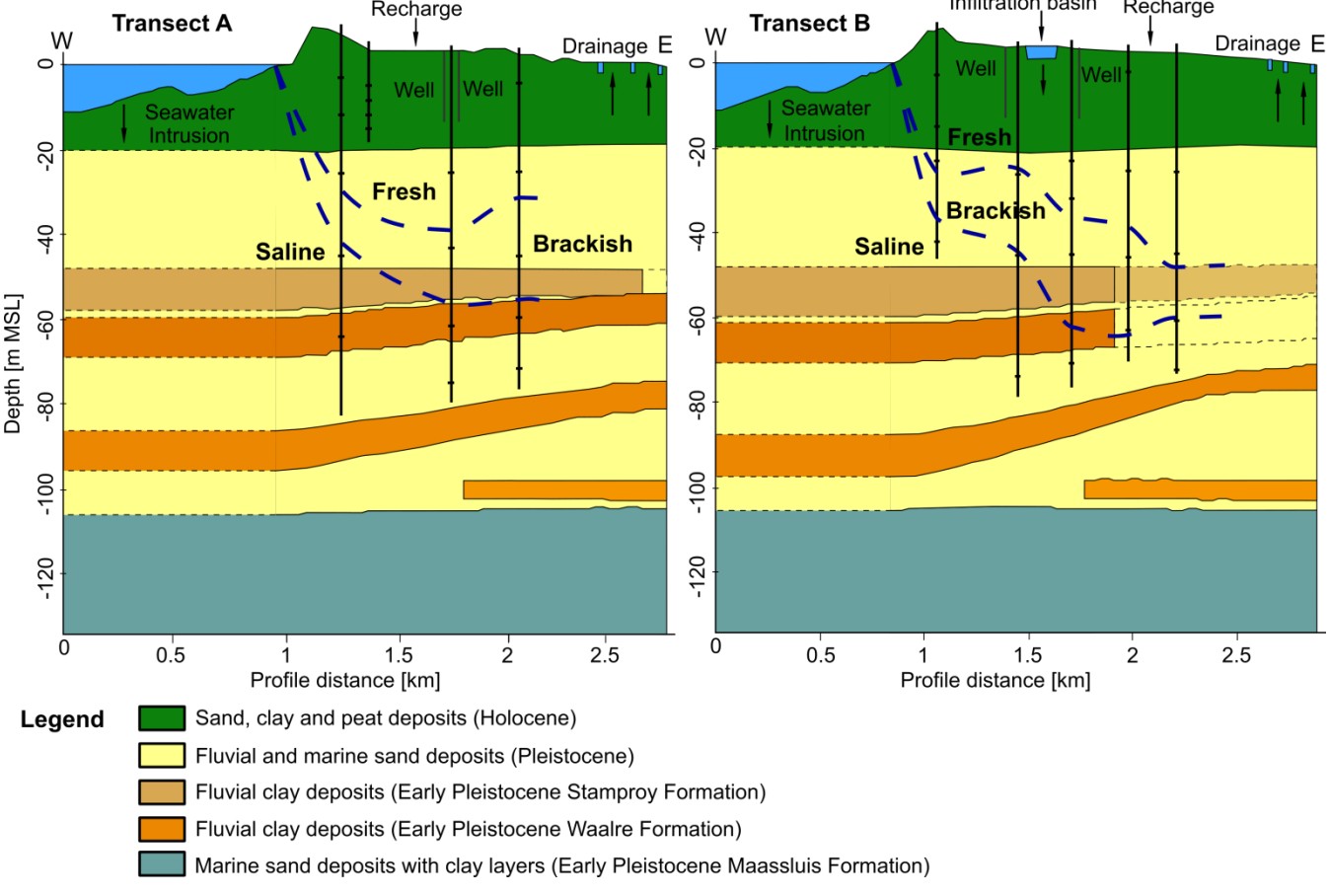

**Fig. 3.** Geological profiles (based on the databases of the Geological Survey of The Netherlands) across the model domain with conceptual fresh-salt water distribution (locations are shown in **Fig. 2**)

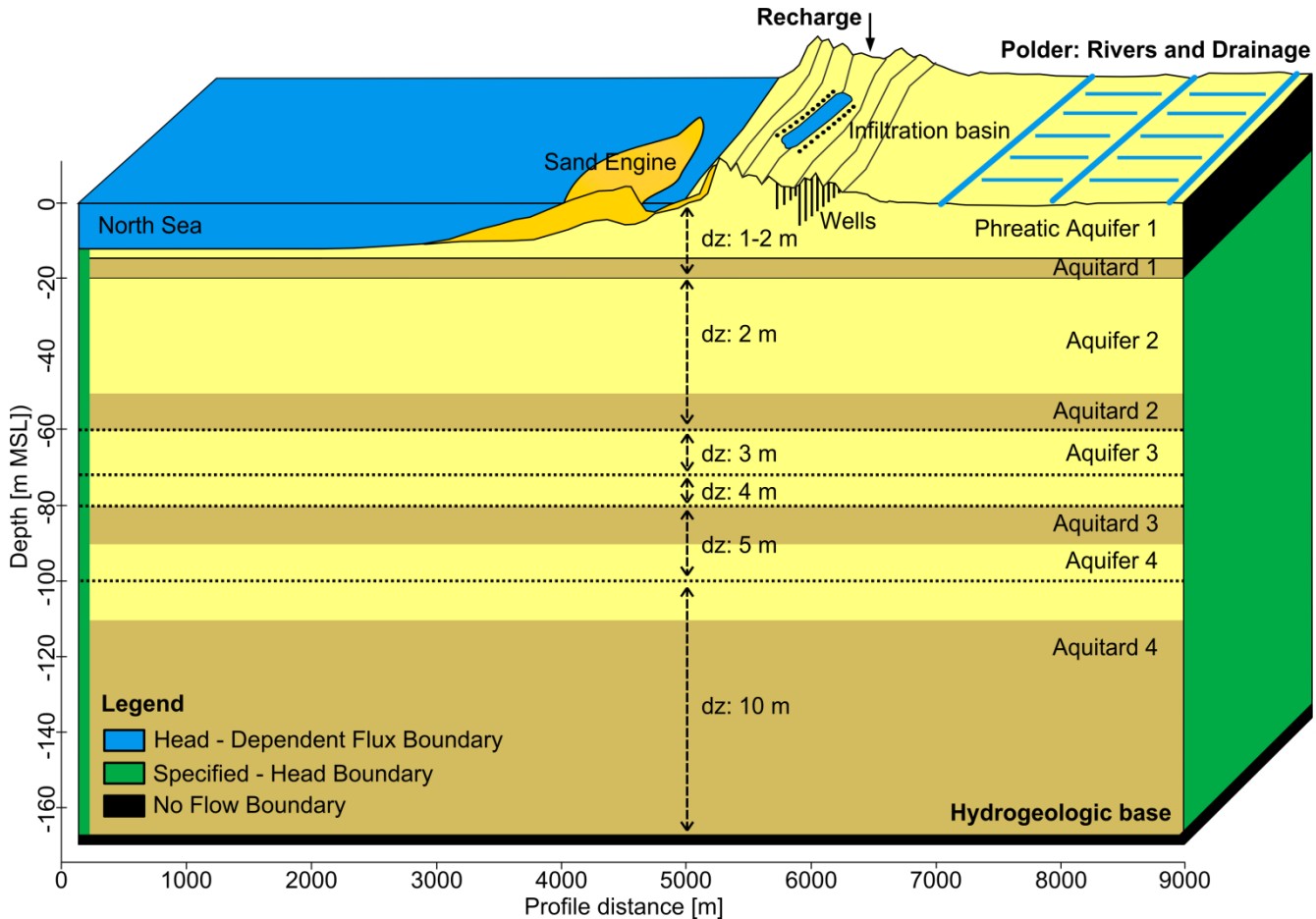

**Fig. 4.** Conceptual representation of a slice of the model with layer thicknesses and boundary conditions

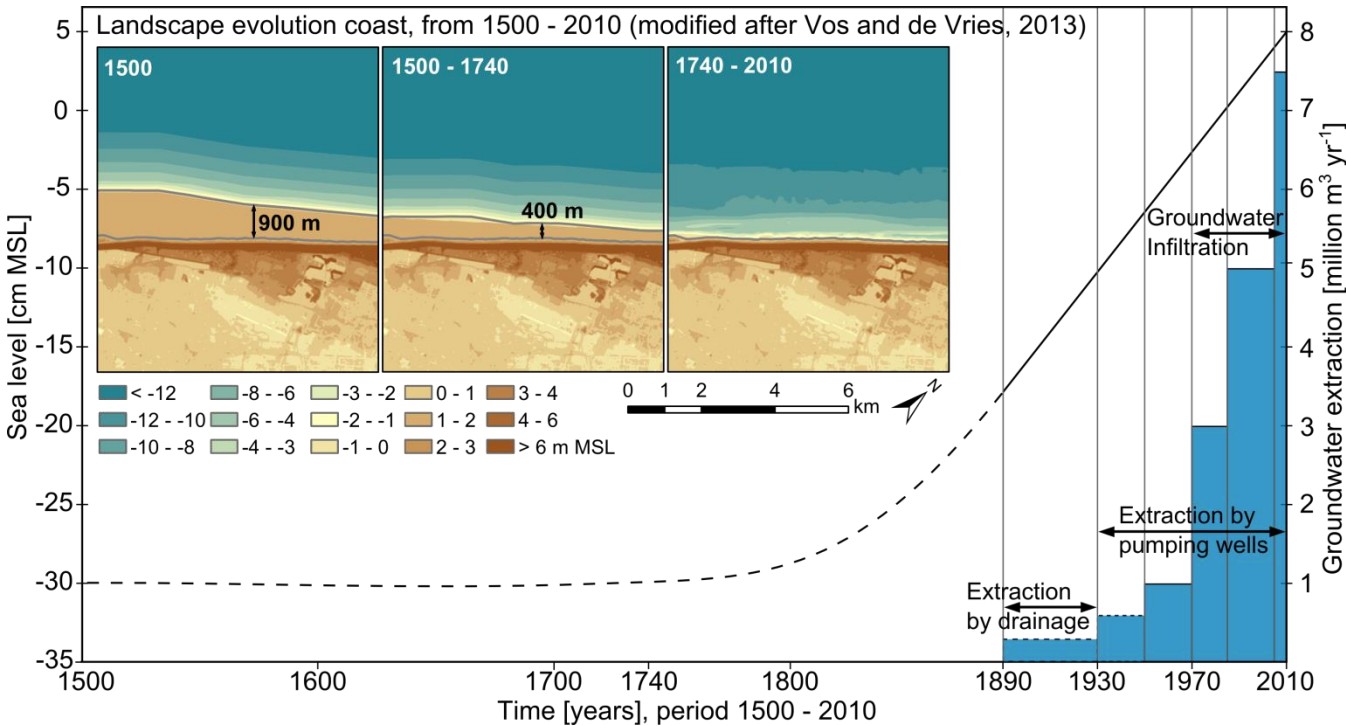

**Fig. 5.** Simulation of historical coastal erosion (based on paleogeographic maps of Vos and de Vries, 2013), sea-level rise (black line) and groundwater extraction (blue) in the period 1810 – 2010; dashed lines indicate estimates, and vertical grey lines refer to stress periods (CBS et al., 2013).

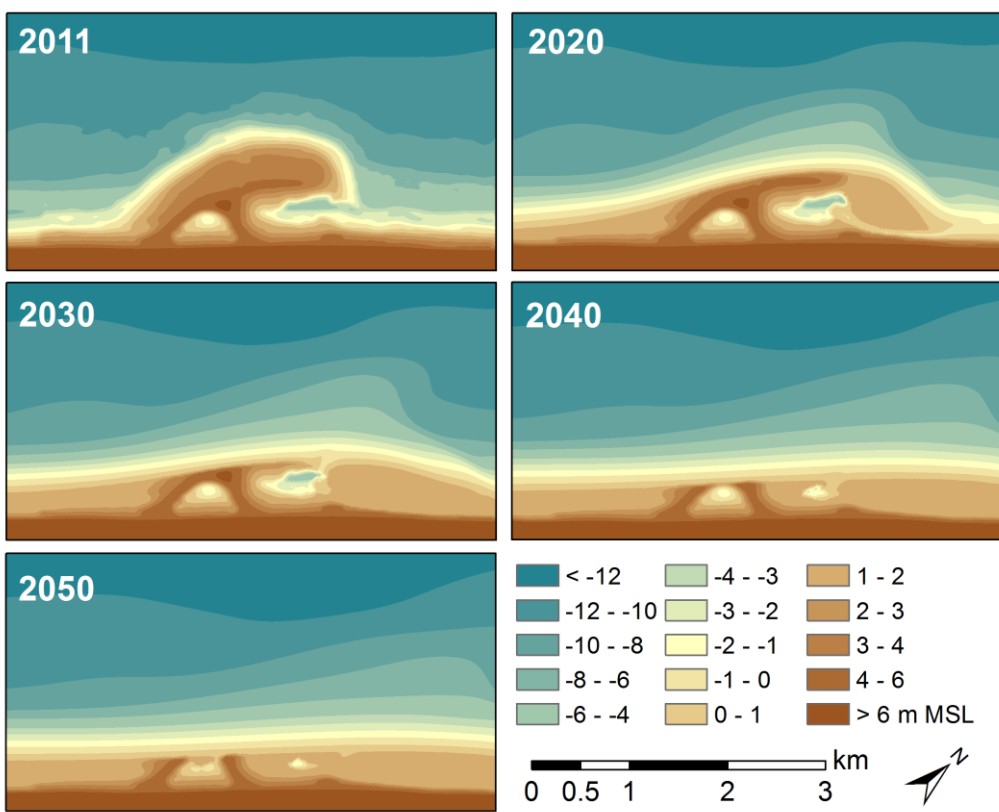

**Fig. 6.** Simulated morphological development of the Sand Engine from 2011 to 2050, illustrated by contour maps with the terrain elevation (m MSL)

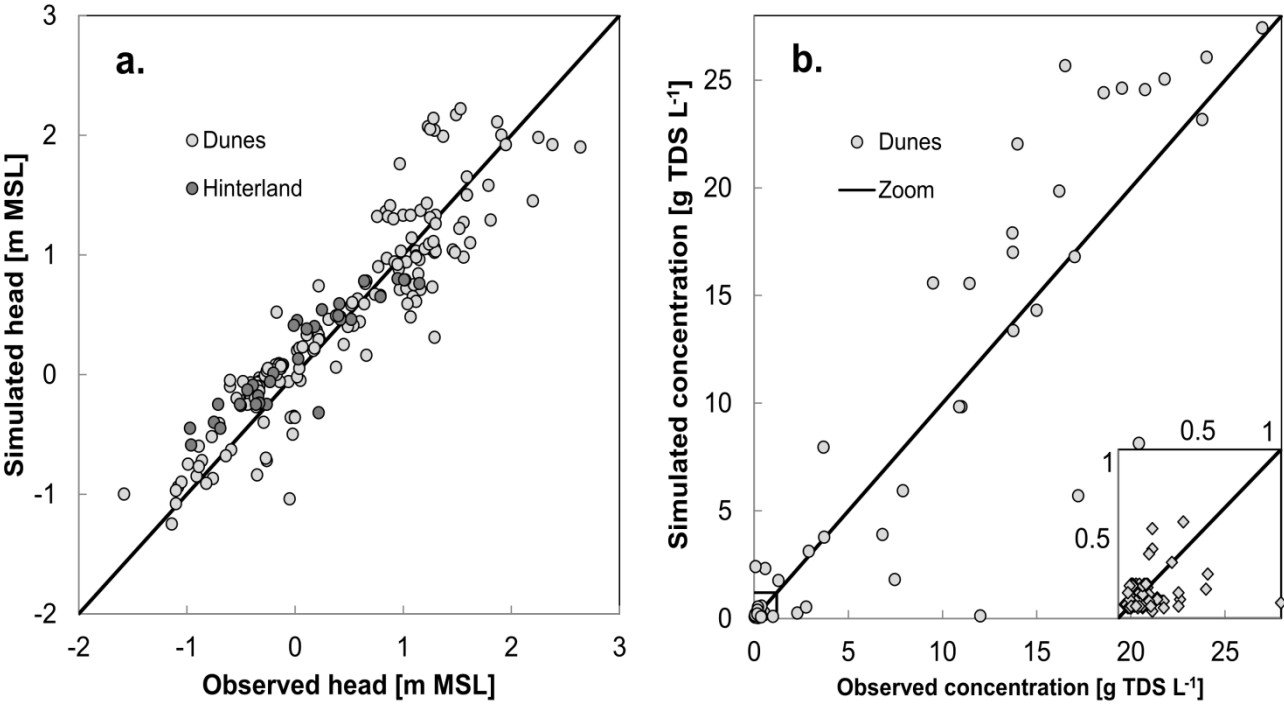

**Fig. 7.** Comparison of observed and simulated heads (a) and concentration (b)

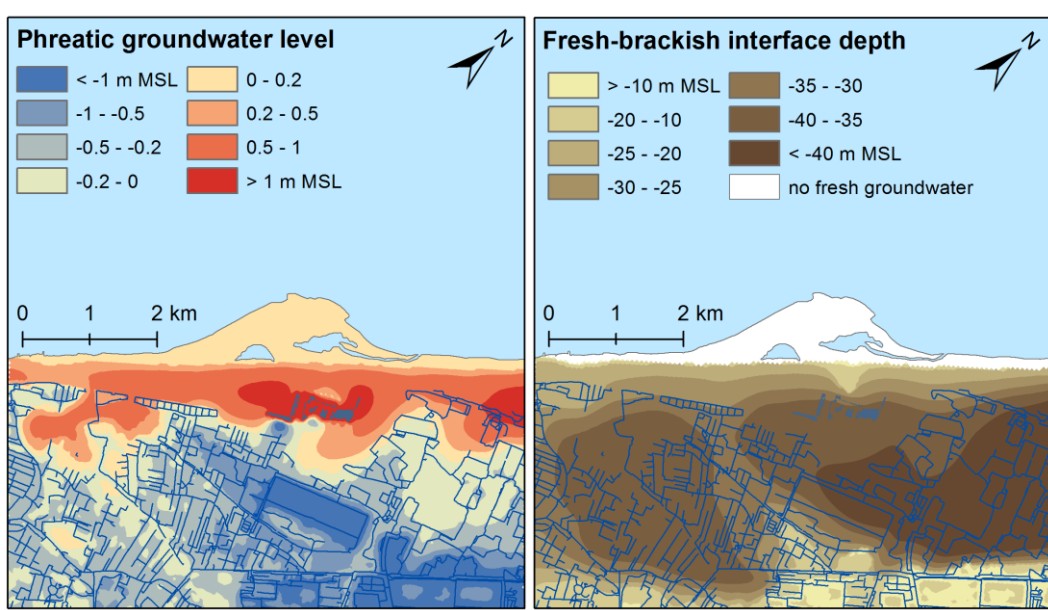

**Fig. 8.** Phreatic groundwater level (left) and fresh-brackish interface depth (right) after calibration, before the construction of the Sand Engine (2010 – 2011).

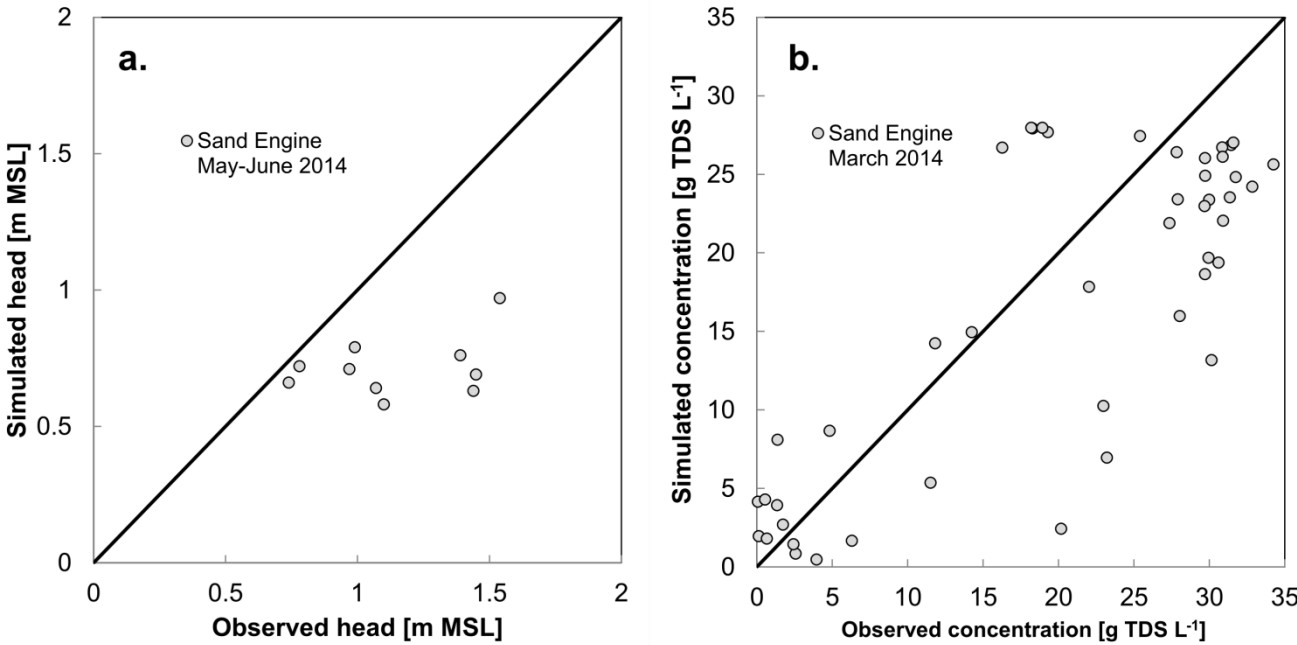

**Fig. 9.** Comparison of (a) average groundwater heads in May-June 2014 and (b) (single) TDS concentrations of soil samples taken between 10 and 14 March 2014 with model simulations in the Sand Engine.

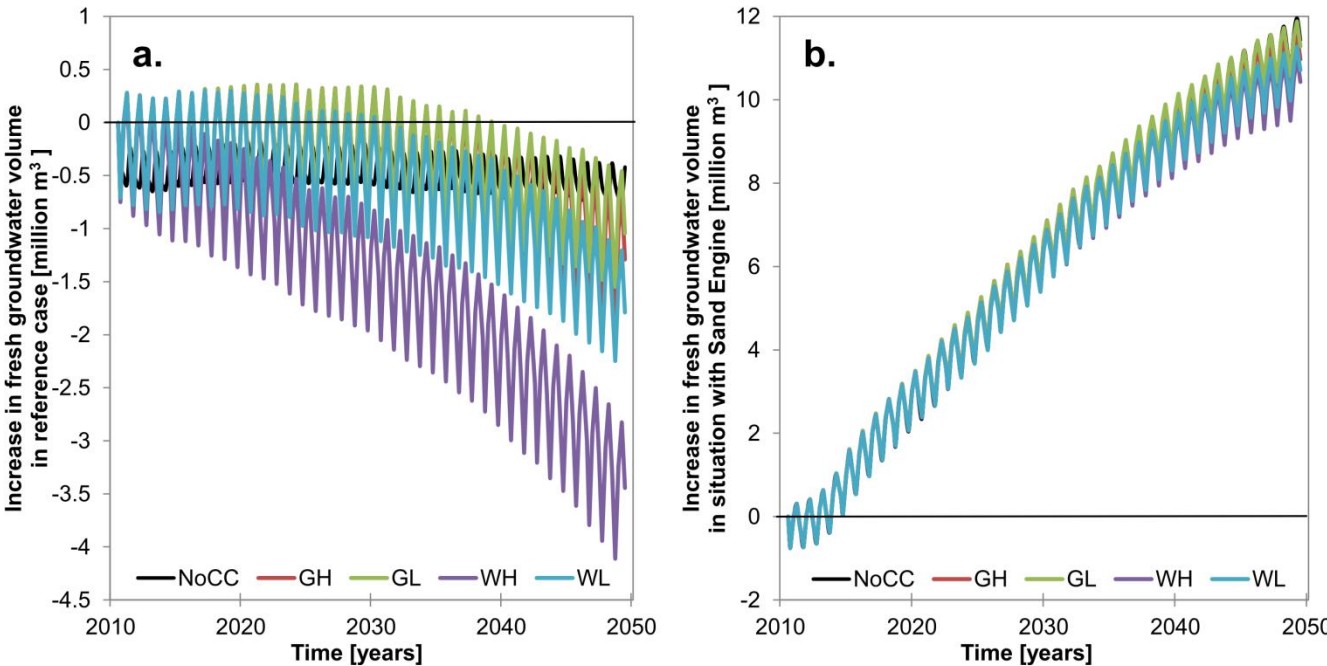

**Fig. 10.** Increase of the volume of fresh groundwater in the situation without Sand Engine (a) and situation with Sand Engine (b) in the period 2011 to 2050, where the legend refers to (climate) scenarios (as mentioned in Table 3)

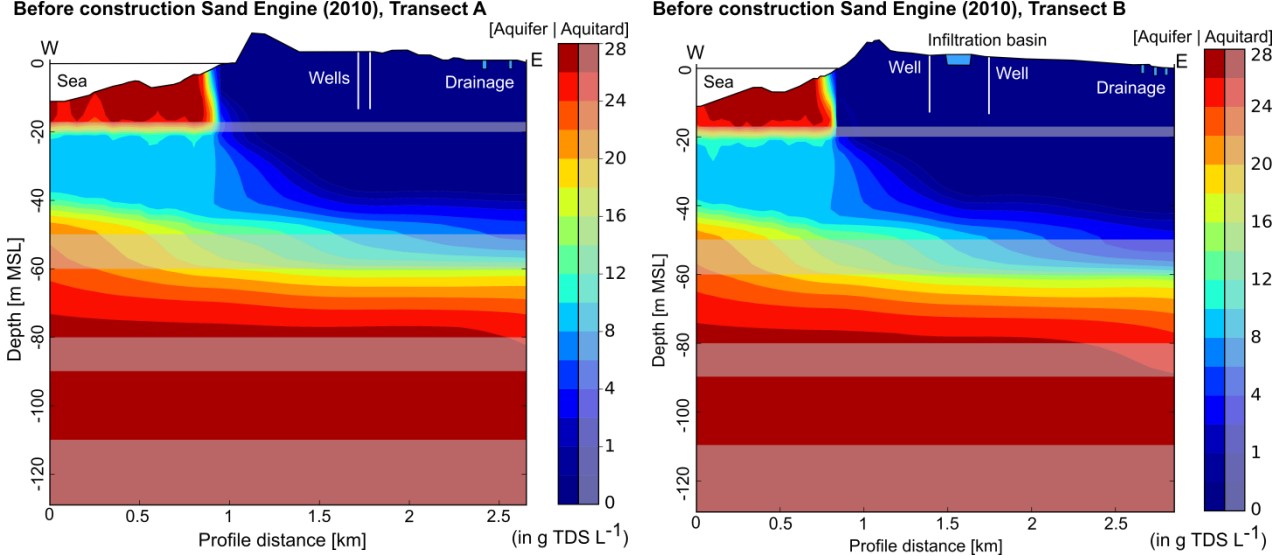

**Fig. 11.** Transects with the simulated groundwater salinity (in g TDS L-1) in 2010 (pre-development Sand Engine), for transect A and B (as shown in Fig. 2 and Fig. 3)

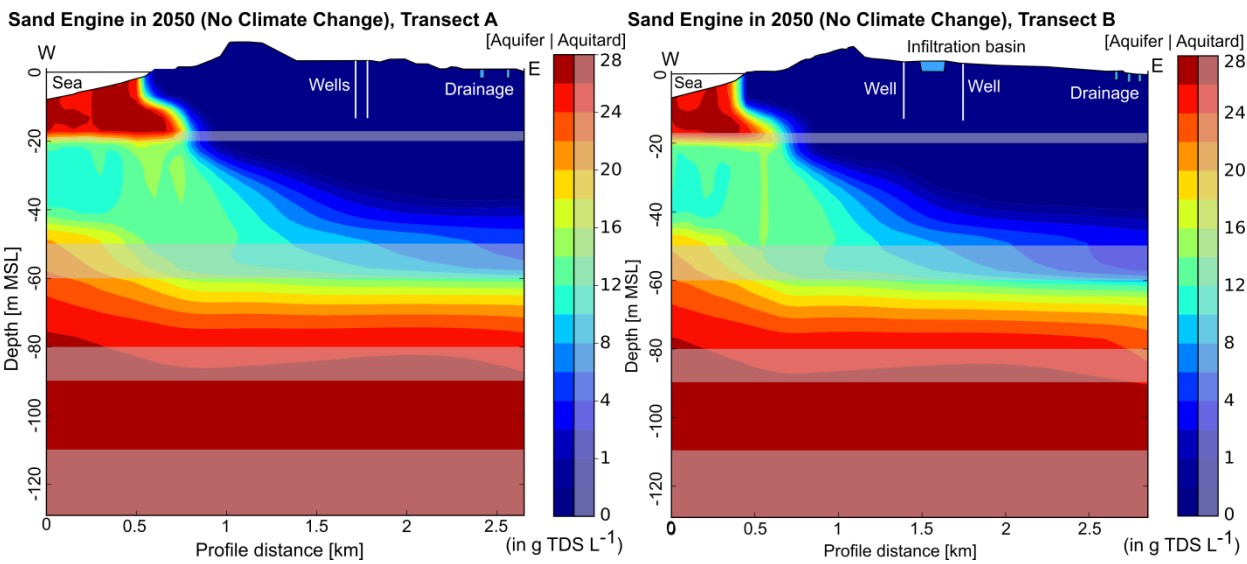

5   **Fig. 12.** Transects with the simulated groundwater salinity (in g TDS L$^{-1}$) in 2050 (including Sand Engine, No Climate Change), for transect A and B (as shown in Fig. 2 and Fig. 3)

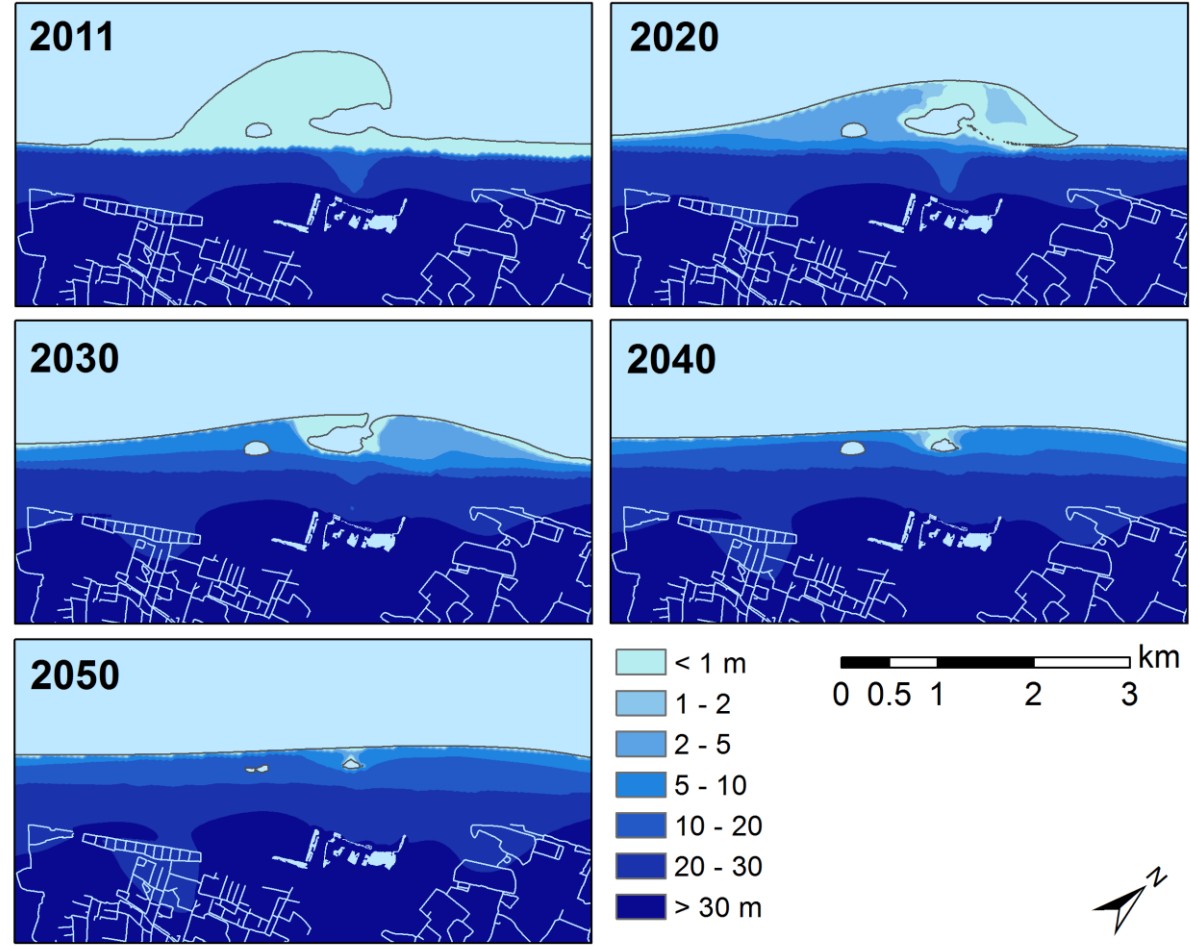

**Fig. 13.** Thickness of fresh groundwater [m] in reference scenario near the Sand Engine from 2011 – 2050

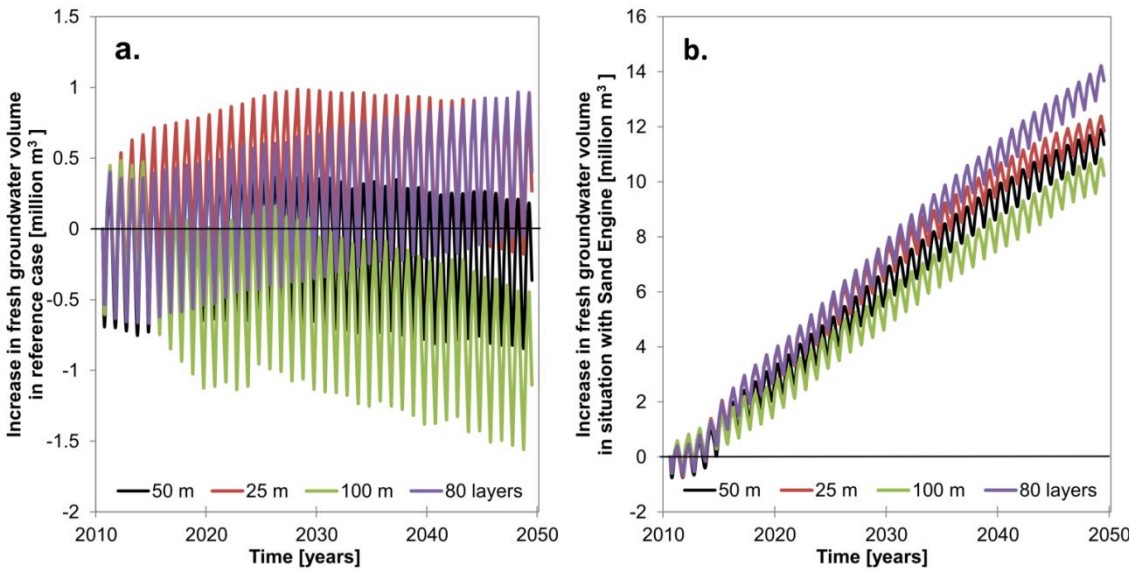

**Fig. A1.** Increase of the volume of fresh groundwater in the situation without Sand Engine (a) and situation with Sand Engine (b) in the period 2011 to 2050, where the legend refers to the four grid discretisation simulations

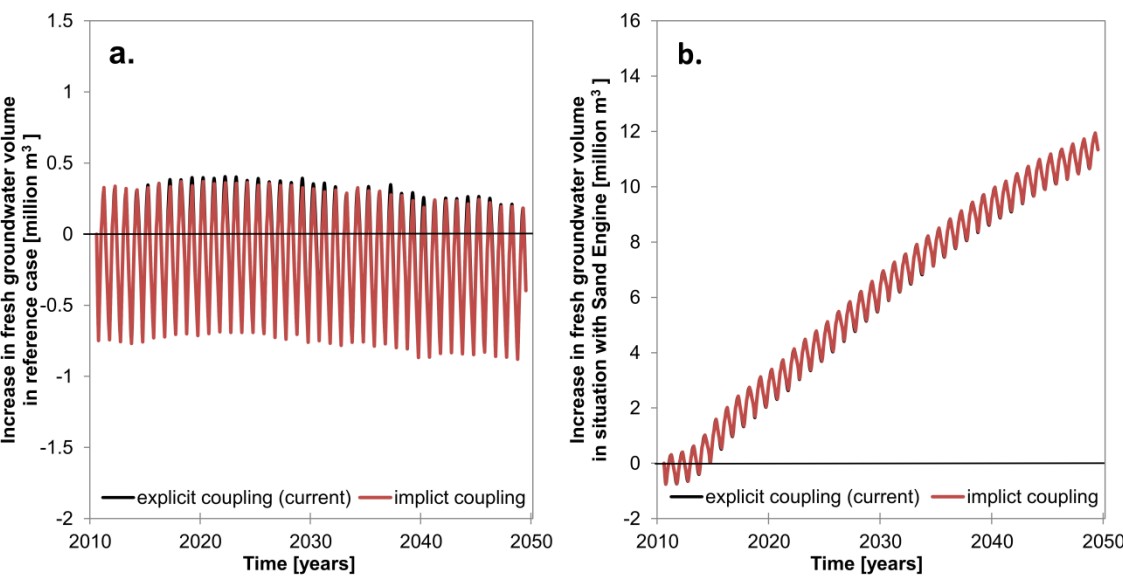

5   **Fig. A2.** Increase of the volume of fresh groundwater in the situation without Sand Engine (a) and situation with Sand Engine (b) in the period 2011 to 2050, where the legend refers to the coupling of flow and transport equations

**Table A1. Grid Convergence Index (GCI) of the simulated increase in the volume of fresh groundwater in 2050 (situation with Sand Engine), for three simulations that contain different spatial grid refinements (S1 to S3).**

| Sim. | Grid size [m] | Layers | Refinement ratio[a] [-] | Relative error[a] [-] | Order of accuracy[a,b] [-] | GCI [%] |
|------|---------------|--------|--------------------------|------------------------|-----------------------------|---------|
| S1 | 25 x 25 | 50 | 1.585 | 0.0224 | 3.36 | 0.76 |
| S2 | 100 x 100 | 50 | 1.585 | 0.1074 | 3.36 | 3.63 |
| S3 | 50 x 50 | 80 | 1.286 | 0.1487 | 3.36[c] | 14.0 |

[a] The refinement ratio, relative error and order or accuracy are parameters that were used to determine the GCI (Roache, 1994), [b] The order of accuracy was estimated with the relative error and refinement ratio of model simulation S1 and S2 (Stern et al., 2001), [c] For model simulation S3 the estimated order of accuracy of simulation S1 and S2 was used to calculate the GCI.