# Peer review of "Fresh groundwater resources in a large sand replenishment"

_Hydrology and Earth System Sciences, 2016_

## Referee Comment (RC1) · Anonymous Referee #1 · 8 Feb 2016

Paper Review for Hydrology and Earth System Sciences

Authors Huizer et al. have submitted the manuscript (ms) entitled "Fresh groundwater resources in a large sand replenishment" to Hydrology and Earth System Sciences.

The authors investigate the effect of local sand replenishment along a Dutch coastal regional on fresh groundwater resources. Effects of climatic change are also included in the study. A 3D variable-density groundwater flow model is constructed and used to predict long-term effects of the added sand. Results indicate that the sand replenishment can both protect the coastline and secure freshwater resources at the same time. It is concluded that sand replenishment can have that combined positive effect along many other low-lying coastal areas worldwide.

The ms is a valuable and novel contribution to coastal flow research. The ms is well

written and organized. Figures are of good quality. The key message is novel and should be made available to the scientific community. However, the manuscript does have room for some substantial improvement that mainly concerns its modeling part. My major points of criticism are given as General Comments, followed by some specific and technical comments that must also be addressed. I recommend acceptance of the manuscript with major revisions.

General Comments

1. P6L1-6. The authors give details on the spatial grid. The last phrases indicate that the authors are aware of spatial discretization problems inherent to variable-density flow simulations. However, it appears that the authors have not conducted a grid convergence test. Does that spatial discretization exclude numerical round-off and truncation errors? The simulations are certainly transient (the authors should say so), so have the authors examined the effect of temporal discretization? What is the time-step size? Which time-stepping scheme is adopted, constant, adaptive, error-controlled? Both the spatial and temporal discretization must be justified. All of this must be clarified in the revised ms version.

2. P6L7-14. Please clearly explain the definition of BCs, this is not clear from Fig. 4. A 2D slice might be helpful. What is a general-head BC? Does the constant-head BC apply to the top of the sea or to the sea floor?

3. P7L31-34 (and other locations). It should be clarified and clearly listed which processes both models simulate, and how they are incorporated. For example, how is coastal erosion incorporated in the groundwater model? While I do understand that Delft3D is a sediment transport model, I do not see that it can also simulate erosion? Please clarify. Also, how was sea-level rise incorporated in the groundwater model? Your model is not a box-type model domain so your beach is actually inclined. As a consequence, more beach surface area is inundated as a result of sea-level rise. This changes the type of BC of beach nodes from Dirichlet to Neumann. How was this

issue dealt with? And also, it appears from P8L27 that tidal activity was simulated by the groundwater model, is this correct? If so, more details on that BC are required: which tidal signal was imposed, how does the time-step size change as a results of tidal activity? How was that tidal BC incorporated on the beach boundary?

4. Section 3.2 Initial Conditions. It is unclear to me which model(s) you run to attain initial conditions. I would believe it is computationally almost impossible to simulate a coupled morphodynamic-groundwater model for 510 years that includes sediment transport, erosion, saltwater intrusion, sea-level rise, tidal activity, submarine freshwater discharge, variable-density groundwater flow, salt transport. That simulation alone would require a very rigorous choice of spatial and temporal discretization, and the small time-step size would very significantly increase the CPU times. Again, listing of processes simulated by which model is obligatory here. Also, it appears from P8L4f that you are simulating the salt distribution at the onset of your actual simulation. This implies that the salt concentration in the North Sea is not at steady state, which requires clarification.

5. How was the newly simulated sand distribution communicated to the groundwater model? Was the spatial grid deformed corresponding to the newly simulated bathymetry? Did you re-mesh the model area? How was the sea-zone represented: high-K zone? Which K does the sea have?

6. Section 4.1 Model Calibration. It must be clarified and justified here that you calibrated on steady-state (recent) values of head and salinity. That calibrated steady-state model is then used to run transient scenarios. Hence, the transient model is, strictly speaking, uncalibrated!

Specific Comments

7. P1L24. Inhabitants are not vulnerable, ecosystems are.

8. P2L6. Are the Netherlands a delta? Either call it "region" or "country, or simply

delete.

9. P2L19f. This is ill-phrased. What you mean is that instead of putting a little bit of sand everywhere, people think about putting a lot of sand on one point in space. The term "small-scale" is misleading here because it is actually a large-scale distribution of sand that is being replaced by point-wise replenishment of sand. This needs to be written in appropriate terms.

10. P2L23. The "surface level including the sea bed level" is simply the bathymetric surface, or even simpler the bathymetry.

11. P4L24. It is unclear which unit the phreatic aquifer is. I am guessing the green unit in Fig. 3? A legend in Fig. 3 would be helpful.

12. P4L34. There is no freshwater lens in your coastal aquifer. To my understanding, freshwater lenses only form below islands and in coastal aquifers under heavy influence of groundwater extraction that pushes the saltwater-freshwater interface upwards forming a lens on the seaside of the pumping wells. Neither is the case here.

13. P5L1f are obvious, delete.

14. P5L10. Swap words: "frequently measured". Also, how frequently?

15. P6L7. Delete "outer" since all boundaries are along the outside. Also, some boundaries are parallel to the coastline, so the first phrase needs rewording. Please indicate the location of your model area in Fig. 2. As is, it is unclear where exactly you are modeling. Is it the rectangle in Fig. 1?

16. P6L15-20. Please indicate in Fig. 4 which is an aquifer and which is an aquitard. All units could simply be named aquifer 1,2,... aquitard 1,2,..., phreatic aquifer etc., and a legend should be given. Also, please put all the parameter values in a table and delete from the text.

17. P6L22f. I do not see the phreatic aquifer nor the two hydrogeological layers in any

figure. This must be clarified.

18. P6L29ff. Do you mean spatially or temporally averaged values? Surely, the simulation is transient, then are all these values constant-in-time and spatially distributed?

19. P7L6. The *linear* relation?

20. P7L27. Delete "method" and "of". Replace "adjustment" by "calibration". How was the model calibration done, manually, PEST? Please clarify. Same for P8L10-14, how did you actually find the values of the finally calibrated parameter?

21. P9L16. "evenly or randomly" is ill-worded.

22. P11L12-24 are Intro material and should be shifted.

23. P13L9. Unclear which "local circumstances" you mean. Either clarify or delete.

24. Fig. 10. What causes the oscillations? Tidal activity? This must be explained and it must be said, which tidal signal is applied. A scale on the time axis is missing, probably 2011-2050? Simulating tidal activity for 40 years would require a very small time-step size. Or did you only consider the lunar cycle in the change of the sea level?

25. Fig. 11 (and corresponding interpretation in text). Did you consider the morphological situation of 2050 as a steady state? What happens after 2050?

26. Literature. References on the effect of tidal activity and storm surges on coastal freshwater resources could be mentioned: Kooi, H., Groen, J., Leijnse, A., 2000. Modes of seawater intrusion during transgressions. Water Resources Research 36, 3581-3589. Violette, S., Boulicot, G., Gorelick, S.M., 2009. Tsunami-induced groundwater salinization in southeastern India. Comptes Rendus Geoscience 341, 339-346. Yang, J., Graf, T., Herold, M., Ptak, T., 2013. Modelling the effects of tides and storm surges on coastal aquifers using a coupled surface-subsurface approach. J. Contam. Hydrol. 149, 61-75. Yang, J., Graf, T., Ptak, T., 2015. Sea level rise and storm surge effects in a coastal heterogeneous aquifer: a 2D modelling study in northern Germany.
[Figure]

Grundwasser 20, 39-51.

Technical Comments

27. P2L4, P2L14, P2L31 (and many other locations in the ms). Please add a comma: "Fortunately,", "Since 2001,", "In September 2011,". I found approximately 30 missing commas.

28. P2L11. "have"

29. P3L1. "800 m into the sea"

30. P3L2. Fig. 2 not 1

31. P3L4. Delete "(local mega-nourishment)", it is now clear.

32. P3L17. Consistently use "variable-density" with "-".

33. P3L23 (and other locations). Replace "scenario's" by "scenarios".

34. P4L26. Delete "grained".

35. P5L6. "long-term".

36. P5L28. "were simulated".

37. P10L12. "similar to the situation".

38. P10L21. "volume of groundwater".

39. P10L22. Replace "lower" by "smaller".

40. P11L3. Replace "with" by "by".

41. P11L5. Replace "pace" by "rate".

42. P13L5. Swap words: "substantially grow".

43. Table 1. Plus the effect of the Sand Engine gives a total of 10 scenarios? Please
clarify.

44. Fig. 7b. Give values of the zoom plot a different symbol to better differentiate.

---

## Referee Comment (RC2) · Anonymous Referee #2 · 9 Feb 2016

General comments The authors state that the volume of replenished sand in their case is "large". Without a comparison to previous nourishments, the reader cannot judge if the volume 21.5 Mill. m$^3$ is indeed large. Please give some figures for previous measures for comparison. The potential negative effects of a mega-nourishment should at least be mentioned briefly. Where does the sand come from? How does the extraction affect currents and wildlife there? What about sandbanks forming downstream which may obstruct shipping? Not sure whether your model cell size is appropriate for the initial steps of freshwater generation in the sand engine, when the freshwater body is still small Why would a wetter winter lead to a lower volume of fresh groundwater (P10, L21-24). Should a wetter winter not lead to more recharge in NW European climate? List references by year of publication, oldest go first (e.g. in Line 20) The manuscript would be a better read after a liberal sprinkling of commas! Not sure about HESS

policy but should non-English sources in the references come with a translation? e.g. Buma (2013), P 14, L5 and others

Specific comments Page 1, Line 20: use spelling "deltas" not"delta's" P. 1, Line 21: usual spelling in English is "Vietnam" P2, L6: not the whole of the Netherlands is a delta, right? People in Friesland and Limburg would probably not agree P2, L10-12: sand nourishment is not only done in the NL, the Germans do it, too, and probably other countries as well P2, L11-12: how often is sand nourishment usually done? Every year, every five, ten, twenty years? P2, L15; replace "must rise" by "rises" P2, L23: Weren't their some presentations on the sand engine at the latest SWIM in Husum? Please cite references if appropriate P2, L28: replace "determined" by "investigated" P2, L31: please replace "shape" by a more appropriate term describing the geometry P3, L1: replace "in" by "into" (twice!) P3, L12: "displacements in seawater intrusion" sounds awkward please rephrase P3, L13: no need to define SGD, delete text inparentheses P3, L17/18: does variable density gw flow not include salt transport? (same for P5, L27) P3, L23 and 24: replace "scenario's" by "scenarios" P4, L5: probably "rainbowing" is the correct spelling?! P4, L10: delete "clean," P4, L13-15: how much groundwater is infiltrated, how much is extracted, how much is locally formed? P4, L24: replace "are" by "were" (same in Line 28) P4, L25-26: an aquifer made up of clay? are you sure? P5, L1: delete comma P5, L10: replace "observed" by "read off" P5, L12-14: these were on-shore in the dunes, right? P5, L15-19: the purpose of these wells remains unclear, are they pumping saline/brackish water as interceptor wells? Are they running continuously? Please specify! P6; L15': add "the" after the second "and" P6; L27/28: here you use m/d while above (L19) you use SI standards (m, s) P6; L29-34: the values chosen for these data should be stated somewhere, maybe in a table P7, L32-33: but HOW were they incorporated? and which ones? in what timescale? P8, L25: why not use "every three months"? P9, L17: add "and" instead of comma P10, L12: add "the" before "situation" P11, L3: replace "with" by "by" P11, L14-19: not sure whether a comparison to island lenses is appropriate here. This is also no conclusion but a introductory note. Maybe better deleted! P11, L31: since you raise the issue: how

many times was the sand engine flooded? Fig. 1: add north arrow Fig. 1: legend for gray scales? Fig. 3: explain formation names, maybe ages or so? Fig. 4: values for general head boundaries? give legend to identify aquitards and aquifers Fig. 5: modified after Vos 2013? Fig. 8: which year is shown? Fig. 10: explain in caption that the labels refer to (climate) scenarios

―――――――――――――――――――――

---

## Referee Comment (RC3) · Anonymous Referee #3 · 26 Feb 2016

This paper describes groundwater modelling of the impact on freshwater resources of a local sand nourishment development off the coast of the Netherlands, called the 'Sand Engine'. The modelling effort includes morphological changes of the 'Sand Engine' caused by wind, currents and tides. The model is loosely calibrated and then used as a predictive tool under different climate scenarios. The paper is very well written and the quality of the figures is very high. Modelling freshwater resources within a moving sand island is interesting and novel. There is an appropriate amount of background detail provided. The technical aspects of the work appear to be sound and the limitations of the modelling effort are well detailed. The conclusion that local sand replenishments can provide both coastal protection and increasing freshwater availability is important and of general interest.

---

## Author Comment (AC1) · 31 Mar 2016

We would like to thank the Referee for the comments, which are highly appreciated.

"This paper describes groundwater modelling of the impact on freshwater resources of a local sand nourishment development off the coast of the Netherlands, called the 'Sand Engine'. The modelling effort includes morphological changes of the 'Sand Engine' caused by wind, currents and tides. The model is loosely calibrated and then used as a predictive tool under different climate scenarios. The paper is very well written and the quality of the figures is very high. Modelling freshwater resources within a moving sand island is interesting and novel. There is an appropriate amount of background detail provided. The technical aspects of the work appear to be sound and the limitations of the modelling effort are well detailed. The conclusion that local sand re-

plenishments can provide both coastal protection and increasing freshwater availability is important and of general interest."

Referee#3 did not submit general or specific comments.

---

## Author Comment (AC2) · 4 Apr 2016

We would like to thank the Referee for the comments, which are highly appreciated. We will improve the raised issues.

General Comments

"1. P6L1-6. The authors give details on the spatial grid. The last phrases indicate that the authors are aware of spatial discretization problems inherent to variable-density flow simulations. However, it appears that the authors have not conducted a grid convergence test. Does that spatial discretization exclude numerical round-off and truncation errors? The simulations are certainly transient (the authors should say so), so have the authors examined the effect of temporal discretization? What is the time-step size? Which time-stepping scheme is adopted, constant, adaptive, error-controlled? Both

the spatial and temporal discretization must be justified. All of this must be clarified in the revised ms version."

We agree with the Referee, and will perform additional simulations for the reference scenario to justify the spatial and temporal discretization: we will perform additional simulation with horizontal resolutions of 25 m and 100 m and one simulation with an increased vertical resolution. The flow simulations were performed with stress periods of 90.25 to 92 days (corresponding to seasons), which were further divided into 3 time steps, resembling approximately one month for each flow time step. The transport simulations were performed implicitly with the Generalized Conjugate Gradient (GCG) solver, and the transport step sizes were model-calculated (DT0 equal to zero) based on a Courant number of 1. Additionally we have used a transport step size multiplier of 1.02. In order to justify the temporal discretization we will perform an explicit transport simulation. The justification of our chosen values and results of the additional simulations will be reported in a supplement; in the paper we will refer to the supplement.

"2. P6L7-14. Please clearly explain the definition of BCs, this is not clear from Fig. 4. A 2D slice might be helpful. What is a general-head BC? Does the constant-head BC apply to the top of the sea or to the sea floor?"

We will change the names of the general-head BC and the constant-head BC in the text and in figure 4 to respectively 'Head-Dependent Flux Boundary' and 'Specified-Head Boundary', in order to clarify the boundary conditions. The constant-head boundary applies to the sea floor, and we will add this to referred section of the paper: P6L10-14.

"3. P7L31-34 (and other locations). It should be clarified and clearly listed which processes both models simulate, and how they are incorporated. For example, how is coastal erosion incorporated in the groundwater model? While I do understand that Delft3D is a sediment transport model, I do not see that it can also simulate erosion? Please clarify. Also, how was sea-level rise incorporated in the groundwater model? Your model is not a box-type model domain so your beach is actually inclined. As a

consequence, more beach surface area is inundated as a result of sea-level rise. This changes the type of BC of beach nodes from Dirichlet to Neumann. How was this issue dealt with? And also, it appears from P8L27 that tidal activity was simulated by the groundwater model, is this correct? If so, more details on that BC are required: which tidal signal was imposed, how does the time-step size change as a results of tidal activity? How was that tidal BC incorporated on the beach boundary?"

To clarify which processes are simulated and incorporated in the groundwater model, we will add a short table listing the processes and the method of incorporation. The processes that are referred to in section P7L31-34 are visualized in Fig. 5. This figure illustrates how these processes are incorporated in the groundwater model. The historical coastal erosion is based on paleogeographic maps by Vos and de Vries (2013), the reference to this source was shown in Fig. 5. We will clarify this in the description of the figure.

The morphological change of the Sand Engine was based on simulations with Delft3D (Lesser et al., 2004), with computations of the hydrodynamics, waves, sediment transport and morphology. We will add references to papers in which this is described: - Mulder, J. P. M. and Tonnon, P. K.: "Sand Engine" : Background and design of a mega-nourishment pilot in the Netherlands, in Proceedings of International Coastal Engineering Conference 32, pp. 1–10, Shanghai, China., 2011. - Tonnon, P. K., van der Werf, J. and Mulder, J. P. M.: Morphological simulations, Environmental Impact Assesment Sand Engine (in Dutch), Deltares, Delft., 2009.

Sea-level rise was incorporated in the groundwater model by performing successive simulations. In each successive simulation or model the level of inundation was determined by comparing the sea-level with the surface level, the boundary conditions were adapted accordingly (P8L24-26). Tidal activity was not simulated by the groundwater model, because this lies beyond the scope of this paper. We will add a brief description and clarification to the paragraph 3.3.

[Figure]

"4. Section 3.2 Initial Conditions. It is unclear to me which model(s) you run to attain initial conditions. I would believe it is computationally almost impossible to simulate a coupled morphodynamic-groundwater model for 510 years that includes sediment transport, erosion, saltwater intrusion, sea-level rise, tidal activity, submarine freshwater discharge, variable-density groundwater flow, salt transport. That simulation alone would require a very rigorous choice of spatial and temporal discretization and the small time-step size would very significantly increase the CPU times. Again, listing of processes simulated by which model is obligatory here. Also, it appears from P8L4f that you are simulating the salt distribution at the onset of your actual simulation. This implies that the salt concentration in the North Sea is not at steady state, which requires clarification."

The morphodynamic simulations were only used for the projection of the morphological evolution of the Sand Engine, the historical coastal erosion was determined by maps in Fig. 5. We will add a listing of the simulated processes to clarify the description, and we will combine this with the request in the previous comment. The 'salt distribution' refers to groundwater salinities, for the salt concentration in the North Sea we have assumed a constant value of 28 g TDS L-1.

"5. How was the newly simulated sand distribution communicated to the groundwater model? Was the spatial grid deformed corresponding to the newly simulated bathymetry? Did you re-mesh the model area? How was the sea-zone represented: high-K zone? Which K does the sea have?"

Yes, the spatial grid was deformed corresponding to the newly simulated bathymetry. And yes, we re-meshed the model area. The whole process was simulated by a series of successive 'deformed or re-meshed' groundwater models, as described briefly in P8L24-26. The sea-zone was excluded from the simulations, and the sea boundary conditions were applied to the seafloor. As mentioned in the response to comment 2, we will add this to the description in the paper (P6L7-14).

"6. Section 4.1 Model Calibration. It must be clarified and justified here that you calibrated on steady-state (recent) values of head and salinity. That calibrated steady-state model is then used to run transient scenarios. Hence, the transient model is, strictly speaking, uncalibrated!"

Yes, we have used averaged values of head and salinity in the calibration. We have made this choice because of the limited availability of long time-series of heads and salinity, and the long-term focus of the paper. We will clarify this in section 4.1, and address this in the discussion P11L27-32.

Specific Comments

"7. P1L24. Inhabitants are not vulnerable, ecosystems are."

We will change 'vulnerable' to 'threatened' and add 'ecosystems' to the line.

"8. P2L6. Are the Netherlands a delta? Either call it "region" or "country, or simply delete."

We agree, we will modify this to 'country'.

"9. P2L19f. This is ill-phrased. What you mean is that instead of putting a little bit of sand everywhere, people think about putting a lot of sand on one point in space. The term "small-scale" is misleading here because it is actually a large-scale distribution of sand that is being replaced by point-wise replenishment of sand. This needs to be written in appropriate terms."

We will replace the term "small-scale" with "frequent large-scale distribution of sand", and "local mega-nourishments" with "concentrated (mega) nourishments".

"10. P2L23. The "surface level including the sea bed level" is simply the bathymetric surface, or even simpler the bathymetry".

We will modify the text to 'surface elevation (including bathymetry)'

"11. P4L24. It is unclear which unit the phreatic aquifer is. I am guessing the green unit in Fig. 3? A legend in Fig. 3 would be helpful."

Yes, the green unit corresponds with the Holocene deposits, and contains the phreatic aquifer and underlying aquitard. We will add a legend to Fig. 3 to clarify the colours.

"12. P4L34. There is no freshwater lens in your coastal aquifer. To my understanding, freshwater lenses only form below islands and in coastal aquifers under heavy influence of groundwater extraction that pushes the saltwater-freshwater interface upwards forming a lens on the seaside of the pumping wells. Neither is the case here."

The term 'freshwater lens' refers to the existence (and development) of fresh groundwater on top of saline groundwater in the coastal aquifer. We will change the name to 'fresh groundwater lens'.

"13. P5L1f are obvious, delete."

We agree and will delete the second part of this sentence.

"14. P5L10. Swap words: "frequently measured". Also, how frequently?"

We will replace this with 'twice every year'

"15. P6L7. Delete "outer" since all boundaries are along the outside. Also, some boundaries are parallel to the coastline, so the first phrase needs rewording. Please indicate the location of your model area in Fig. 2. As is, it is unclear where exactly you are modeling. Is it the rectangle in Fig. 1?"

We will delete "outer", and yes the model boundaries are shown as the rectangle in Fig. 1. We will modify and clarify this in the ms.

"16. P6L15-20. Please indicate in Fig. 4 which is an aquifer and which is an aquitard. All units could simply be named aquifer 1,2,. . . aquitard 1,2,. . ., phreatic aquifer etc., and a legend should be given. Also, please put all the parameter values in a table and delete from the text."

We will indicate in Fig. 4 which layers are aquitards and which layers are aquifers and put the parameter values in a separate table.

"17. P6L22f. I do not see the phreatic aquifer nor the two hydrogeological layers in any figure. This must be clarified."

We will modify the text in this paragraph to clarify the subdivision of the phreatic aquifer.

"18. P6L29ff. Do you mean spatially or temporally averaged values? Surely, the simulation is transient, then are all these values constant-in-time and spatially distributed?"

In this section we mean temporally averaged values, for example seasonally averaged for recharge and yearly averaged values for surface water levels. These model parameters were all spatially distributed. Yes, the simulation is transient, and these values are therefore constant per season for recharge, and constant for the whole simulation for surface water levels. We will modify the text to clarify the adopted methodology.

"19. P7L6. The *linear* relation?"

Yes, we used the linear relation between chloride and TDS. We will add 'linear' to the text.

"20. P7L27. Delete "method" and "of". Replace "adjustment" by "calibration". How was the model calibration done, manually, PEST? Please clarify. Same for P8L10-14, how did you actually find the values of the finally calibrated parameter?"

We will change the sentence according to the suggestions. The calibration was performed manually, and we will clarify this in the text. We manually adjusted the values of a selection of model parameters (as mentioned in P8L10-14) within realistic ranges to attain the best calibration fit. The adjustments were made from an initial best guess of the values. We clarify the text in both paragraphs to clarify the adopted methodology.

"21. P9L16. "evenly or randomly" is ill-worded."

We agree, evenly suggest a regular pattern in contrast with randomly. We will change

the phrasing into "well-distributed".

"22. P11L12-24 are Intro material and should be shifted."

We will shift the paragraph to the introduction, in paragraph 1.3 on P3L21.

"23. P13L9. Unclear which "local circumstances" you mean. Either clarify or delete."

We will change "local circumstances" to "local hydrogeological conditions".

"24. Fig. 10. What causes the oscillations? Tidal activity? This must be explained and it must be said, which tidal signal is applied. A scale on the time axis is missing, probably 2011-2050? Simulating tidal activity for 40 years would require a very small time-step size. Or did you only consider the lunar cycle in the change of the sea level?"

The oscillations are caused by seasonal changes in recharge (winter, spring, summer, autumn). Tidal activity was not included in the simulations. Both figures to contain a time axis with labels from 2010 to 2050, however this may be difficult to read in figure 10a. We will adapt the position of time-axis.

"25. Fig. 11 (and corresponding interpretation in text). Did you consider the morphological situation of 2050 as a steady state? What happens after 2050?"

No, the morphological situation will continue to change after 2050. However we have limited the morphological simulations to this period, because the main effects of the Sand Engine on fresh groundwater resources become apparent in this period.

"26. Literature. References on the effect of tidal activity and storm surges on coastal freshwater resources could be mentioned: - Kooi, H., Groen, J., Leijnse, A., 2000. Modes of seawater intrusion during transgressions. Water Resources Research 36, 3581-3589. - Violette, S., Boulicot, G., Gorelick, S.M., 2009. Tsunami-induced groundwater salinization in southeastern India. Comptes Rendus Geoscience 341, 339-346. - Yang, J., Graf, T., Herold, M., Ptak, T., 2013. Modelling the effects of tides and storm surges on coastal aquifers using a coupled surface-subsurface approach. J. Contam.

[Figure]

Hydrol. 149, 61-75. - Yang, J., Graf, T., Ptak, T., 2015. Sea level rise and storm surge effects in a coastal heterogeneous aquifer: a 2D modelling study in northern Germany. Grundwasser 20, 39-51."

We will consider the mentioning of some of these references, and were appropriate.

Technical Comments

"27. P2L4, P2L14, P2L31 (and many other locations in the ms). Please add a comma: "Fortunately,", "Since 2001,", "In September 2011,". I found approximately 30 missing commas."

We will add these commas, and will check the ms for other missing commas.

"28. P2L11. "have""

We will change the sentence to '. . . application of sand nourishments has . . .'

"29. P3L1. "800 m into the sea""

We will correct this in the ms.

"30. P3L2. Fig. 2 not 1"

We think both figures are appropriate, however Fig. 1 contains images of the morphological change

"31. P3L4. Delete "(local mega-nourishment)", it is now clear."

We agree, and will delete this.

"32. P3L17. Consistently use "variable-density" with "-". 33. P3L23 (and other locations). Replace "scenario's" by "scenarios"."

We will correct this in the ms.

"34. P4L26. Delete "grained"."

We will add hyphens to fine and medium to make clear that these words refer to grain size

"35. P5L6. "long-term"."

We will correct this in the ms.

"36. P5L28. "were simulated"."

We will delete "and salt transport", which makes the original "was simulated" correct.

"37. P10L12. "similar to the situation". 38. P10L21. "volume of groundwater". 39. P10L22. Replace "lower" by "smaller". 40. P11L3. Replace "with" by "by". 41. P11L5. Replace "pace" by "rate". 42. P13L5. Swap words: "substantially grow"."

We will correct this in the ms.

"43. Table 1. Plus the effect of the Sand Engine gives a total of 10 scenarios? Please clarify."

Yes, this table only contains the climate change scenarios. We will delete the words 'model', and change to 'climate (change) scenario'.

"44. Fig. 7b. Give values of the zoom plot a different symbol to better differentiate."

We will change the symbols in the zoom plot.

---

## Author Comment (AC3) · 4 Apr 2016

We would like to thank the Referee for the comments, which are highly appreciated. We will improve the raised issues.

**General Comments**

"The authors state that the volume of replenished sand in their case is "large". Without a comparison to previous nourishments, the reader cannot judge if the volume 21.5 Mill. m3 is indeed large. Please give some figures for previous measures for comparison. "

We agree that the statement "large" is subjective and will remove this statement from P2L25. In the beginning of the paragraph (P2L14-16) we have mentioned the "traditional" nourishment volume of 12 million m3, which is (on average) applied yearly along

the entire Dutch coast.

"The potential negative effects of a mega-nourishment should at least be mentioned briefly. Where does the sand come from? How does the extraction affect currents and wildlife there? What about sandbanks forming downstream which may obstruct shipping?"

We think that these issues are not relevant for the paper, and in addition, many of these issues are still under investigation within the large Nature Coast programme. Possible advantages were only mentioned as motivations for the creation of the Sand Engine. However, we will adapt the text to remove the impression that a mega-nourishment will predominantly have positive effect.

"Not sure whether your model cell size is appropriate for the initial steps of freshwater generation in the sand engine, when the freshwater body is still small. "

We agree with the Referee, and will perform additional simulations for the reference scenario to justify the spatial and temporal discretization. In order to justify the spatial discretization, additional simulations will be performed with horizontal resolutions of 25 m and 100 m and one simulation with an increased vertical resolution. The results of the additional simulations will be reported in a supplement; in the paper we will refer to the supplement.

"Why would a wetter winter lead to a lower volume of fresh groundwater (P10, L21-24). Should a wetter winter not lead to more recharge in NW European climate?"

Yes, a wetter winter would lead to a higher volume of fresh groundwater, however the text is comparing climate scenarios. The climate scenarios with a high response will lead to higher volume of fresh groundwater in comparison with the climate scenario's with a weak response. Overall the groundwater recharge increases more in the milder climate scenarios. We will adapt these sentences to clarify the intention.

"List references by year of publication, oldest go first (e.g. in Line 20) "

HESSD
In the manuscript preparation guidelines for authors it is stated that 'In terms of in-text citations, the order can be based on relevance, as well as chronological or alphabetical listing, depending on the author's preference.' We have chosen to list in-text citations by alphabetical order.

"The manuscript would be a better read after a liberal sprinkling of commas!"

We will check the manuscript for readability, and add commas were possible.

"Not sure about HESS C1-English sources in the references come with a translation? e.g. Buma (2013), P 14, L5 and others"

We will translate all Non-English titles to English; P14L5, P14L30, P15L1, P16L1-2, P17L12-13, P18L13-15, P18L16, P18L26-27, P19L17, P19L18-19, P19L26-27, P19L29, P20L15.

**Specific Comments**

"Page 1, Line 20: use spelling "deltas" not" delta's" P. 1, Line 21: usual spelling in English is "Vietnam""

We will change this in the ms.

"P2, L6: not the whole of the Netherlands is a delta, right? People in Friesland and Limburg would probably not agree"

Yes, we will modify the text from 'delta' to 'country'

"P2, L10-12: sand nourishment is not only done in the NL, the Germans do it, too, and probably other countries as well"

This is correct, and we have briefly addressed this in L4-5, however not specificly.

"P2, L11-12: how often is sand nourishment usually done? Every year, every five, ten, twenty years?"

We will change the line in the ms.
"P2, L15; replace "must rise" by "rises""

We will correct this in the ms as "is to rise"

"P2, L23: Weren't their some presentations on the sand engine at the latest SWIM in Husum? Please cite references if appropriate"

Yes, the preliminary results that are described in this paper were presented at the SWIM in Husum.

"P2, L28: replace "determined" by "investigated""

We will correct this in the ms.

"P2, L31: please replace "shape" by a more appropriate term describing the geometry"

We will adapt the line with more appropriate terms; 'retreat of outer perimeter' and 'increase alongshore extent'

"P3, L1: replace "in" by "into" (twice!) "

We will correct this in the ms.

"P3, L12: "displacements in seawater intrusion" sounds awkward please rephrase"

We will adapt this to "to dynamic changes in seawater ...."

"P3, L13: no need to define SGD, delete text in parentheses"

We agree, and will delete definition.

"P3, L17/18: does variable density gw flow not include salt transport? (same for P5, L27)"

We agree, in the absence of other species there is no need to include salt transport here.

"P3, L23 and 24: replace "scenario's" by "scenarios" P4, L5: probably "rainbowing" is
the correct spelling?!"

We will correct this in the ms.

"P4, L10: delete "clean," "

We will change this and rephrase the sentence in the ms.

"P4, L13-15: how much groundwater is infiltrated, how much is extracted, how much is locally formed?"

We will add this information to the paragraph.

"P4, L24: replace "are" by "were" (same in Line 28) P4, L25-26: an aquifer made up of clay? are you sure? P5, L1: delete comma P5, L10: replace "observed" by "read off" "

We will correct this in the ms.

"P5, L12-14: these were on-shore in the dunes, right?"

Yes, in the dunes and in some in the hinterland (urban area, polders). We will add 'onshore'

"P5, L15-19: the purpose of these wells remains unclear, are they pumping saline/brackish water as interceptor wells? Are they running continuously? Please specify! "

Yes, these wells serve as interceptor wells; they control the groundwater level to avoid any possible negative impact of the nourishments. We will clarify this in the paragraph.

"P6; L15: add "the" after the second "and""

We will correct this in the ms.

"P6; L27/28: here you use m/d while above (L19) you use SI standards (m, s) "

We have used the most common and appropriate unit for each model parameter.
"P6; L29-34: the values chosen for these data should be stated somewhere, maybe in a table"

We will transfer the values of the model parameter to a table.

"P7, L32-33: but HOW were they incorporated? and which ones? in what timescale? "

The processes (coastal erosion, sea-level rise, and expansions of groundwater drainage and extractions) that are referred to in section P7L29-34 are visualized in Fig. 5. This figure illustrates how these processes are incorporated in the groundwater model. The historical coastal erosion is based on paleogeographic maps by Vos and de Vries (2013), the reference to this source was shown in Fig. 5. We will clarify this in the figure caption, and adapt the sentences in section P7L29-34 to clarify the methodology.

"P8, L25: why not use "every three months"?"

We will change "quarter' to "every three months", because this is more explicit.

"P9, L17: add "and" instead of comma P10, L12: add "the" before "situation" P11, L3: replace "with" by "by" "

We will correct this in the ms.

"P11, L14-19: not sure whether a comparison to island lenses is appropriate here. This is also no conclusion but a introductory note. Maybe better deleted!"

We will move this section toward the introduction (paragraph 1.3), and will delete the addition (L17-19) to reduce the focus on island lenses in this section.

"P11, L31: since you raise the issue: how many times was the sand engine flooded? "

Only certain areas of the Sand Engine have been flooded. Until now there have been two 'major' storms in 2011 and 2013 that lead to large inundations, and several 'minor' storms leading to less extensive inundations. We will this information to the paragraph.
"Fig. 1: add north arrow Fig. 1: legend for gray scales?"

We will add this to the figure.

"Fig. 3: explain formation names, maybe ages or so? "

We will add a legend to the figure with some information about the formations (age, lithology)

"Fig. 4: values for general head boundaries? give legend to identify aquitards and aquifers"

The values of the general head boundaries were taken from a previous model simulation of the southwest of the Netherlands (Oude Essink et al., 2010). We will add a legend to the figure, and to identify which layers are aquitards and aquifers.

"Fig. 5: modified after Vos 2013?"

We will correct this in the figure.

"Fig. 8: which year is shown?"

This is the year after calibration, before the construction of the Sand Engine (2010 - 2011). We will add this to the figure caption.

"Fig. 10: explain in caption that the labels refer to (climate) scenarios"

We will explain this in the figure caption.

---

## Author Response (AR1)

**Reply on review of our manuscript HESS-2016-5 "Fresh groundwater resources in a large sand replenishment" by Anonymous Referee #1, #2, #3**

Dear Editor,

We would like to thank the reviewers for their effort reviewing our manuscript and their valuable comments.

In the subsequent pages we have explained point by point how we dealt with their comments, arranged from Anonymous Referee #1 to #3. The original reviewer comments are presented in bold italic, and our response is presented in normal text. The marked up manuscript version is added to at the end of this document.

Sincerely,

Sebastian Huizer, on behalf of all authors

**AnonymousReferee#1**

*The authors investigate the effect of local sand replenishment along a Dutch coastal regional on fresh groundwater resources. Effects of climatic change are also included in the study. A 3D variable-density groundwater flow model is constructed and used to predict long-term effects of the added sand. Results indicate that the sand replenishment can both protect the coastline and secure freshwater resources at the same time. It is concluded that sand replenishment can have that combined positive effect along many other low-lying coastal areas worldwide.*

*The ms is a valuable and novel contribution to coastal flow research. The ms is well written and organized. Figures are of good quality. The key message is novel and should be made available to the scientific community. However, the manuscript does have room for some substantial improvement that mainly concerns its modeling part. My major points of criticism are given as General Comments, followed by some specific and technical comments that must also be addressed. I recommend acceptance of the manuscript with major revisions.*

We would like to thank the Referee for the comments, which are highly appreciated.

**General Comments**

*1. P6L1-6. The authors give details on the spatial grid. The last phrases indicate that the authors are aware of spatial discretization problems inherent to variable-density flow simulations. However, it appears that the authors have not conducted a grid convergence test. Does that spatial discretization exclude numerical round-off and truncation errors?*

*The simulations are certainly transient (the authors should say so), so have the authors examined the effect of temporal discretization? What is the time-step size? Which time-stepping scheme is adopted, constant, adaptive, error-controlled? Both the spatial and temporal discretization must be justified. All of this must be clarified in the revised ms version.*

We agree with the Referee, and have performed additional simulations for the reference scenario to justify the spatial and temporal discretization: we have performed additional simulation with horizontal resolutions of 25 m and 100 m and one simulation with an increased vertical resolution.

The flow simulations were performed with stress periods of 90.25 to 92 days (corresponding to seasons), which were further divided into 3 time steps, resembling approximately one month for each flow time step. The transport simulations were performed implicitly with the Generalized Conjugate Gradient (GCG) solver, and the flow and transport equations were explicitly coupled. The transport step sizes were model-calculated based on a Courant number of 1. Additionally we have used a transport step size multiplier of 1.02. In order to justify the temporal discretization we have performed an additional simulation with an implicit coupling of flow and transport equations.

The justification of our chosen values and results of the additional simulations is reported in Appendix A.

*2. P6L7-14. Please clearly explain the definition of BCs, this is not clear from Fig. 4. A 2D slice might be helpful. What is a general-head BC? Does the constant-head BC apply to the top of the sea or to the sea floor?*

We have changed the names of the general-head BC and the constant-head BC in the text and in Figure 4 to respectively 'Head-Dependent Flux Boundary' and 'Specified-Head Boundary', in order to clarify the boundary conditions. The constant-head boundary applies to the sea floor, and we have added this to referred section of the paper (section 3.1).

*3. P7L31-34 (and other locations). It should be clarified and clearly listed which processes both models simulate, and how they are incorporated.*

*For example, how is coastal erosion incorporated in the groundwater model? While I do understand that Delft3D is a sediment transport model, I do not see that it can also simulate erosion? Please clarify. Also, how was sea-level rise incorporated in the groundwater model? Your model is not a box-type model domain so your beach is actually inclined. As a consequence, more beach surface area is inundated as a result of sea-level rise. This changes the type of BC of beach nodes from Dirichlet to Neumann. How was this issue dealt with?*

*And also, it appears from P8L27 that tidal activity was simulated by the groundwater model, is this correct? If so, more details on that BC are required: which tidal signal was imposed, how does the time-step size change as a results of tidal activity? How was that tidal BC incorporated on the beach boundary?*

To clarify which processes are simulated and incorporated in the groundwater model, we have added a short table listing the processes and the method of incorporation (Table 2).

The processes that are referred to in section P7L31-34 are visualized in Fig. 5. This figure illustrates how these processes are incorporated in the groundwater model. The historical coastal erosion is based on paleogeographic maps by Vos and de Vries (2013), the reference to this source was shown in Fig. 5. We have clarified this in the description of the figure.

The morphological change of the Sand Engine was based on simulations with Delft3D (Lesser et al., 2004), with computations of the hydrodynamics, waves, sediment transport and morphology. We have added references to papers in which this is described:
- Mulder, J. P. M. and Tonnon, P. K.: "Sand Engine" : Background and design of a mega-nourishment pilot in the Netherlands, in Proc. of International Coastal Engineering Conference 32, pp. 1–10, Shanghai, China., 2011.
- Tonnon, P. K., van der Werf, J. and Mulder, J. P. M.: Morphological simulations, Environmental Impact Assessment Sand Engine (in Dutch), Deltares, Delft, 2009.

Sea-level rise was incorporated in the groundwater model by performing successive simulations. In each successive simulation or model the level of inundation was determined by comparing the sea-level with the surface level, the boundary conditions were adapted accordingly. Tidal activity was not simulated by the groundwater model, because this lies beyond the scope of this paper. We have changed and clarified the description in section 3.3.

*4. Section 3.2 Initial Conditions. It is unclear to me which model(s) you run to attain initial conditions. I would believe it is computationally almost impossible to simulate a coupled morphodynamic-groundwater model for 510 years that includes sediment transport, erosion, saltwater intrusion, sea-level rise, tidal activity, submarine freshwater discharge, variable-density groundwater flow, salt transport. That simulation alone would require a very rigorous choice of spatial and temporal discretization and the small time-step size would very significantly increase the CPU times. Again, listing of processes simulated by which model is obligatory here.*

*Also, it appears from P8L4f that you are simulating the salt distribution at the onset of your actual simulation. This implies that the salt concentration in the North Sea is not at steady state, which requires clarification.*

The morphodynamic simulations were only used for the projection of the morphological evolution of the Sand Engine, the historical coastal erosion was determined by maps in Fig. 5. We have added a list of the simulated processes to clarify the description (Table 2), and have combined this with the request in the previous comment. The 'salt distribution' refers to groundwater salinities, for the salt concentration in the North Sea we have assumed a constant value of 28 g TDS $L^{-1}$.

**5. How was the newly simulated sand distribution communicated to the groundwater model? Was the spatial grid deformed corresponding to the newly simulated bathymetry? Did you re-mesh the model area? How was the sea-zone represented: high-K zone? Which K does the sea have?**

Yes, the spatial grid was deformed, corresponding to the newly simulated bathymetry. And yes, we have re-meshed the model area. The whole process was simulated by a series of successive 'deformed or re-meshed' groundwater models, as described briefly in section 3.3. The sea-zone was excluded from the simulations, and the sea boundary conditions were applied to the seafloor. As mentioned in the response to comment 2, we have added this to the description in the paper (section 3.1).

**6. Section 4.1 Model Calibration. It must be clarified and justified here that you calibrated on steady-state (recent) values of head and salinity. That calibrated steady-state model is then used to run transient scenarios. Hence, the transient model is, strictly speaking, uncalibrated!**

Yes, we have used averaged values of head and salinity in the calibration. We have made this choice because of the limited availability of long time-series of heads and salinity, and the long-term focus of the paper. We have clarified this in section 4.1.

**Specific Comments**

**7. P1L24. Inhabitants are not vulnerable, ecosystems are.**

We have changed 'vulnerable' to 'threatened' and have added 'ecosystems' to the line.

**8. P2L6. Are the Netherlands a delta? Either call it "region" or "country, or simply delete.**

We agree, we have modified this to 'country'.

**9. P2L19f. This is ill-phrased. What you mean is that instead of putting a little bit of sand everywhere, people think about putting a lot of sand on one point in space. The term "small-scale" is misleading here because it is actually a large-scale distribution of sand that is being replaced by point-wise replenishment of sand. This needs to be written in appropriate terms.**

We have replaced the term "regular small-scale" with "large-scale distribution of sand",
and the term "local mega-nourishments" with "concentrated (mega) nourishments".

**10. P2L23. The "surface level including the sea bed level" is simply the bathymetric surface, or even simpler the bathymetry.**

We have modified the text to 'surface elevation (including bathymetry)'

**11. P4L24. It is unclear which unit the phreatic aquifer is. I am guessing the green unit in Fig. 3?**
**A legend in Fig. 3 would be helpful.**

Yes, the green unit corresponds with the Holocene deposits, and contains the phreatic aquifer and underlying aquitard. We have added a legend to Fig. 3 to clarify the colours.

*12. P4L34. There is no freshwater lens in your coastal aquifer. To my understanding, freshwater lenses only form below islands and in coastal aquifers under heavy influence of groundwater extraction that pushes the saltwater-freshwater interface upwards forming a lens on the seaside of the pumping wells. Neither is the case here.*

The term 'freshwater lens' refers to the existence (and development) of fresh groundwater on top of saline groundwater in the coastal aquifer. We have changed the name to 'fresh groundwater lens'.

*13. P5L1f are obvious, delete.*

We agree and have deleted the second part of this sentence.

*14. P5L10. Swap words: "frequently measured". Also, how frequently?*

We have replaced this with 'twice every year'

*15. P6L7. Delete "outer" since all boundaries are along the outside. Also, some boundaries are parallel to the coastline, so the first phrase needs rewording. Please indicate the location of your model area in Fig. 2. As is, it is unclear where exactly you are modeling. Is it the rectangle in Fig. 1?*

We have deleted "outer", and yes the model boundaries are shown as a black rectangle in Fig. 1.
We have modified and clarified this in the ms (section 3.1).

*16. P6L15-20. Please indicate in Fig. 4 which is an aquifer and which is an aquitard. All units could simply be named aquifer 1,2,. . . aquitard 1,2,. . ., phreatic aquifer etc., and a legend should be given. Also, please put all the parameter values in a table and delete from the text.*

We have indicated in Fig. 4 which layers are aquitards and which layers are aquifers, and incorporated the parameter values in Table 1.

*17. P6L22f. I do not see the phreatic aquifer nor the two hydrogeological layers in any figure. This must be clarified.*

We have modified the text in this paragraph to clarify the subdivision of the phreatic aquifer (section 3.1).

*18. P6L29ff. Do you mean spatially or temporally averaged values? Surely, the simulation is transient, then are all these values constant-in-time and spatially distributed?*

In this section we mean temporally averaged values, for example seasonally averaged for recharge and yearly averaged values for surface water levels. These model parameters were all spatially distributed. Yes, the simulation is transient, and these values are therefore constant per season for recharge, and constant for the whole simulation for surface water levels. We have modified the text to clarify the adopted methodology (section 3.1).

*19. P7L6. The \*linear\* relation?*

Yes, we have used the linear relation between chloride and TDS. We have added 'linear' to the text.

*20. P7L27. Delete "method" and "of". Replace "adjustment" by "calibration". How was the model calibration done, manually, PEST? Please clarify. Same for P8L10-14, how did you actually find the values of the finally calibrated parameter?*

We have changed the sentence according to the suggestions. The calibration was performed manually, and we have clarified this in the text (section 3.2). We manually adjusted the values of a selection of model parameters (as mentioned in P8L10-14) within realistic ranges to attain the best calibration fit. The adjustments were made from an initial best guess of the values. We clarify the text in both paragraphs to clarify the adopted methodology (section 3.2).

| **21. P9L16. "evenly or randomly" is ill-worded.** |
|---|
| We agree, evenly suggests a regular pattern in contrast with randomly.
We have changed the phrasing into "well-distributed". |

| **22. P11L12-24 are Intro material and should be shifted.** |
|---|
| We have shifted the paragraph to the introduction, in section 1.3. |

| **23. P13L9. Unclear which "local circumstances" you mean. Either clarify or delete.** |
|---|
| We have changed "local circumstances" to "local hydrogeological conditions". |

| **24. Fig. 10. What causes the oscillations? Tidal activity? This must be explained and it must be said, which tidal signal is applied. A scale on the time axis is missing, probably 2011-2050? Simulating tidal activity for 40 years would require a very small time-step size. Or did you only consider the lunar cycle in the change of the sea level?** |
|---|
| The oscillations are caused by seasonal changes in recharge (winter, spring, summer, autumn). Tidal activity was not included in the simulations. Both figures to contain a time axis with labels from 2010 to 2050, however this may be difficult to read in figure 10a. We have adapted the position of time-axis (Fig. 10.) |

| **25. Fig. 11 (and corresponding interpretation in text). Did you consider the morphological situation of 2050 as a steady state? What happens after 2050?** |
|---|
| No, the morphological situation will continue to change after 2050. However we have limited the morphological simulations to this period, because the main effects of the Sand Engine on fresh groundwater resources become apparent in this period. |

| **26. Literature. References on the effect of tidal activity and storm surges on coastal freshwater resources could be mentioned:**
▪ *Kooi, H., Groen, J., Leijnse, A., 2000. Modes of seawater intrusion during transgressions. Water Resources Research 36, 3581-3589.*
▪ *Violette, S., Boulicot, G., Gorelick, S.M., 2009. Tsunami-induced ground-water salinization in southeastern India. Comptes Rendus Geoscience 341, 339-346.*
▪ *Yang, J., Graf, T., Herold, M., Ptak, T., 2013. Modelling the effects of tides and storm surges on coastal aquifers using a coupled surface-subsurface approach. J. Contam. Hydrol. 149, 61-75.*
▪ *Yang, J., Graf, T., Ptak, T., 2015. Sea level rise and storm surge effects in a coastal heterogeneous aquifer: a 2D modelling study in northern Germany. Grundwasser 20, 39-51.* |
|---|
| We have added references in the ms to Kooi et.al. (2000), Yang et.al. (2013) and Yang et.al. (2015). |

**Technical Comments**

| **27. P2L4, P2L14, P2L31 (and many other locations in the ms). Please add a comma: "Fortunately,", "Since 2001,", "In September 2011,". I found approximately 30 missing commas.** |
|---|
| We have added these commas, and have checked the ms for other missing commas. |

| **28. P2L11. "have"** |
|---|
| We have changed the sentence to '… application of sand nourishments has …' |

| **29. P3L1. "800 m into the sea"** |
|---|
| We have corrected this in the ms. |

| |
|---|
| *30. P3L2. Fig. 2 not 1* |
| We think both figures are appropriate, however Fig. 1 contains images of the morphological change |
| *31. P3L4. Delete "(local mega-nourishment)", it is now clear.* |
| We agree, and have delete this. |
| *32. P3L17. Consistently use "variable-density" with "-".* |
| *33. P3L23 (and other locations). Replace "scenario's" by "scenarios".* |
| We have corrected this in the ms. |
| *34. P4L26. Delete "grained".* |
| We have added hyphens to fine and medium to make clear that these words refer to grain size |
| *35. P5L6. "long-term".* |
| We have corrected this in the ms. |
| *36. P5L28. "were simulated".* |
| We have deleted "and salt transport", which makes the original "was simulated" correct. |
| *37. P10L12. "similar to the situation".* |
| *38. P10L21. "volume of groundwater".* |
| *39. P10L22. Replace "lower" by "smaller".* |
| *40. P11L3. Replace "with" by "by".* |
| *41. P11L5. Replace "pace" by "rate".* |
| *42. P13L5. Swap words: "substantially grow".* |
| We have corrected this in the ms. |
| *43. Table 1.  Plus the effect of the Sand Engine gives a total of 10 scenarios?  Please clarify.* |
| Yes, this table only contains the climate change scenarios.  We have deleted the words 'model', and change to 'climate (change) scenario'. |
| *44. Fig. 7b. Give values of the zoom plot a different symbol to better differentiate.* |
| We have changed the symbols in the zoom plot. |

**AnonymousReferee#2**

We would like to thank the Referee for the comments, which are highly appreciated.

**General Comments**
* * *
*The authors state that the volume of replenished sand in their case is "large". Without a comparison to previous nourishments, the reader cannot judge if the volume 21.5 Mill. m3 is indeed large. Please give some figures for previous measures for comparison.*

We agree that the statement "large" is subjective and have removed this statement. In the beginning of the paragraph (section 1.2) we have mentioned the 'traditional' nourishment volume of 12 million $m^3$, which is (on average) applied yearly along the entire Dutch coast.
* * *
*The potential negative effects of a mega-nourishment should at least be mentioned briefly. Where does the sand come from? How does the extraction affect currents and wildlife there? What about sandbanks forming downstream which may obstruct shipping?*

We think that these issues are not relevant for this paper, and many of these issues are still under investigation within the Nature Coast project. Possible advantages were only mentioned as motivation for the creation of the Sand Engine. We have adapted the text to remove the impression that a mega-nourishment will predominantly have positive effects.
* * *
*Not sure whether your model cell size is appropriate for the initial steps of freshwater generation in the sand engine, when the freshwater body is still small.*

We agree with the Referee, and have performed additional simulations for the reference scenario to justify the spatial and temporal discretization. In order to justify the spatial discretization, additional simulations were performed with horizontal resolutions of 25 m and 100 m and one simulation with an increased vertical resolution. The results of the additional simulations are reported in Appendix A.
* * *
*Why would a wetter winter lead to a lower volume of fresh groundwater (P10, L21-24). Should a wetter winter not lead to more recharge in NW European climate?*

Yes, a wetter winter would lead to a higher volume of fresh groundwater. However, the wetter winter in climate scenarios $G_H$ and $W_H$ coincides with a drier summer (comparing climate scenario $G_H$ with $G_L$ and $W_H$ with $W_L$). The increase in the volume of groundwater recharge in the winter season ('wetter winters') is smaller than the decrease of the volume of groundwater recharge in the summer season ('drier summers'). Overall the groundwater recharge increases more in the milder climate scenarios. We have adapted these sentences to clarify the intention (section 4.2).
* * *
*List references by year of publication, oldest go first (e.g. in Line 20)*

In the manuscript preparation guidelines for authors it is stated that 'In terms of in-text citations, the order can be based on relevance, as well as chronological or alphabetical listing, depending on the author's preference.' We have chosen to list in-test citations by alphabetical order.
* * *
| |
|---|
| *The manuscript would be a better read after a liberal sprinkling of commas!* |
| We have checked the manuscript for readability, and have added commas were possible. |
| *Not sure about HESS C1-English sources in the references come with a translation?*
*e.g. Buma (2013), P 14, L5 and others* |
| We have translated all Non-English titles to English. |

**Specific Comments**

| |
|---|
| *Page 1, Line 20: use spelling "deltas" not" delta's"*
*P. 1, Line 21: usual spelling in English is "Vietnam"* |
| We have changed this in the ms. |
| *P2, L6: not the whole of the Netherlands is a delta, right? People in Friesland and Limburg would probably not agree* |
| Yes, we have modified the text from 'delta' to 'country' |
| *P2, L10-12: sand nourishment is not only done in the NL, the Germans do it, too, and probably other countries as well* |
| True, this was addressed in section 1.1. However not specifically. |
| *P2, L11-12: how often is sand nourishment usually done? Every year, every five, ten, twenty years?* |
| We have changed the line in the ms. |
| *P2, L15; replace "must rise" by "rises"* |
| We have corrected this in the ms as "is to rise" |
| *P2, L23: Weren't their some presentations on the sand engine at the latest SWIM in Husum? Please cite references if appropriate* |
| Yes, the preliminary results that are described in this paper were presented at the SWIM in Husum. |
| *P2, L28: replace "determined" by "investigated"* |
| We have corrected this in the ms. |
| *P2, L31: please replace "shape" by a more appropriate term describing the geometry* |
| We have adapted the line with more appropriate terms; 'retreat of outer perimeter' and 'increase alongshore extent' |
| *P3, L1: replace "in" by "into" (twice!)* |
| We have corrected this in the ms. |
| *P3, L12: "displacements in seawater intrusion" sounds awkward please rephrase* |
| We have adapted this to "to dynamic changes in seawater …" |
| *P3, L13: no need to define SGD, delete text in parentheses* |
| We agree, and have deleted the definition. |

| |
|---|
| *P3, L17/18: does variable density gw flow not include salt transport? (same for P5, L27)* |
| We agree, in the absence of other species there is no need to include salt transport here. |
| *P3, L23 and 24: replace "scenario's" by "scenarios"*
*P4, L5: probably "rainbowing" is the correct spelling?!* |
| We have corrected this in the ms. |
| *P4, L10: delete "clean,"* |
| We have changed this and rephrase the sentence in the ms. |
| *P4, L13-15: how much groundwater is infiltrated, how much is extracted, how much is locally formed?* |
| We have added this information to the paragraph. |
| *P4, L24: replace "are" by "were" (same in Line 28)*
*P4, L25-26: an aquifer made up of clay? are you sure?*
*P5, L1: delete comma*
*P5, L10: replace "observed" by "read off"* |
| We have corrected this in the ms. |
| *P5, L12-14: these were on-shore in the dunes, right?* |
| Yes, in the dunes and in some in the hinterland (urban area, polders). We have added 'onshore' |
| *P5, L15-19: the purpose of these wells remains unclear, are they pumping saline/brackish water as interceptor wells? Are they running continuously? Please specify!* |
| Yes, these wells serve as interceptor wells; they control the groundwater level to avoid any possible negative impact of the nourishments. We have clarified this in the paragraph. |
| *P6; L15: add "the" after the second "and"* |
| We have corrected this in the ms. |
| *P6; L27/28: here you use m/d while above (L19) you use SI standards (m, s)* |
| We have used the most common and appropriate unit for each model parameter. |
| *P6; L29-34: the values chosen for these data should be stated somewhere, maybe in a table* |
| We have transferred the values of the model parameter to a table (Table 1). |
| *P7, L32-33: but HOW were they incorporated? and which ones? in what timescale?* |
| The processes (coastal erosion, sea-level rise, and expansions of groundwater drainage and extractions) are visualized in Fig. 5, and the source and method of incorporation of the processes is described in Table 2. We have clarified this in the description of the figure and added a reference in the sentence. |
| *P8, L25: why not use "every three months"?* |
| We have changed "quarter' to "every three months", because this is more explicit. |
| *P9, L17: add "and" instead of comma*
*P10, L12: add "the" before "situation"*
*P11, L3: replace "with" by "by"* |
| We have corrected this in the ms. |

| |
|---|
| *P11, L14-19:  not sure whether a comparison to island lenses is appropriate here.  This is also no conclusion but a introductory note.  Maybe better deleted!* |
| We have moved this section toward the introduction (paragraph 1.3), and have deleted the addition (L17-19) to reduce the focus on island lenses in this section. |
| *P11, L31:  since you raise the issue:  how many times was the sand engine flooded?* |
| Only certain areas of the Sand Engine have been flooded. Until now there have been two 'major' storms in 2011 and 2013 that lead to large inundations, and several 'minor' storms leading to less extensive inundations. We have added this information to section 5. |

| |
|---|
| *Fig. 1:  add north arrow*
*Fig. 1:  legend for gray scales?* |
| We have added a north arrow and legend to the figure. |
| *Fig. 3:  explain formation names, maybe ages or so?* |
| We have added a legend to the figure with some information about the formations (age, lithology) |
| *Fig. 4:  values for general head boundaries?  give legend to identify aquitards and aquifers* |
| The values of the general head boundaries were taken from a previous model simulation of the southwest of the Netherlands (Oude Essink et al., 2010). We have added a legend to the figure, in order to clarify the boundaries, and we have added labels to the aquitards and aquifers. |
| *Fig. 5: modified after Vos 2013?* |
| We have corrected this in the figure. |
| *Fig. 8: which year is shown?* |
| This is the year after calibration, before the construction of the Sand Engine (2010 – 2011).
We have added this to the figure caption. |
| *Fig. 10: explain in caption that the labels refer to (climate) scenarios* |
| We have explained this in the figure caption. |

**AnonymousReferee#3**

**This paper describes groundwater modelling of the impact on freshwater resources of a local sand nourishment development off the coast of the Netherlands, called the 'Sand Engine'. The modelling effort includes morphological changes of the 'Sand Engine' caused by wind, currents and tides. The model is loosely calibrated and then used as a predictive tool under different climate scenarios. The paper is very well written and the quality of the figures is very high. Modelling freshwater resources within a moving sand island is interesting and novel. There is an appropriate amount of background detail provided. The technical aspects of the work appear to be sound and the limitations of the modelling effort are well detailed. The conclusion that local sand replenishments can provide both coastal protection and increasing freshwater availability is important and of general interest.**

We would like to thank the Referee for the comments, which are highly appreciated.

Referee#3 did not submit general or specific comments.

[revised manuscript text omitted]
 et al., 2015; Mahmoodzadeh et al., 2014). In addition, studies have shown that an increase in the frequency of storm

~~surge overwash will exacerbate the salinization of coastal aquifers, although the freshwater lens is generally able to recover over time (Holding et al., 2015; Terry and Falkland, 2010). In relation to the morphological dynamics of coastal regions, studies have shown that the erosion and accretion of sand can lead to substantial changes within the beach foredune area (Bakker et al., 2012; Keijsers et al., 2014b), and that climate change might exacerbate coastal erosion (FitzGerald et al., 2008; Zhang et al., 2004). Morphological developments in coastal areas and islands can therefore have a substantial effect on existing and future fresh groundwater resources. Coastal management strategies that compensate, limit, or counteract coastal erosion or seawater intrusion may therefore help to protect or expand fresh groundwater resources.~~

[revised manuscript text omitted]

**6 Conclusions**

Local mega-nourishments such as the Sand Engine might become an effective solution for the threats that many low-lying coastal regions face, and with this study we have shown that fresh groundwater resources can grow substantially grow within the lifespan of the nourishment. The results in this study show that for the Sand Engine, the construction of a mega-nourishment can lead to increase of fresh groundwater of approximately 0.3 to 0.5 million m$^3$ per year. However, the increase in fresh groundwater resources in a mega-nourishment is highly dependent on the shape and location of the mega-nourishment, the precipitation surplus, the frequency and intensity of storm surges, and the local circumstanceslocal hydrogeological conditions. Therefore dependent on the design and location of the mega-nourishment this may provide an opportunity to combine coastal protection with the protection of fresh groundwater resources. This study also demonstrated that, with relatively simple modifications, a changing morphology can easily be modelled with a variable-density groundwater model such as SEAWAT.

**Appendix A: Grid convergence test**

In order to justify that the chosen spatial and temporal discretisation is adequate for reliable numerical quantifications of the potential effect of the Sand Engine on fresh groundwater resources, we have executed a grid convergence test for period 2011 to 2050 (with and without Sand Engine). Current numerical simulations were performed with: a horizontal grid size of 50 m, 50 layers with a variable thickness from 1 (upper layers) to 10 m (lower layers), and stress periods of 90.25 to 92 days (corresponding to seasons). The transport simulations were performed implicitly with the Generalized Conjugate Gradient (GCG) solver, and the transport step sizes were model-calculated based on a Courant number of 1. For the coupling of the flow and transport we have used the explicit approach. In the grid convergence test we have performed one additional with an implicit (iterative with a density criterion of 0.2 kg m$^{-3}$) to test the temporal discretisation. The spatial discretisation was tested with three additional numerical simulations with higher and lower spatial resolutions: one simulation with a horizontal grid size of 25 m, one simulation with a horizontal grid size of 100 m, and one simulation with the same horizontal grid size of 50 m and an increased vertical resolution of the upper layers. In the upper part of the model, up to a depth of -50 m MSL, the layer thicknesses were lowered with 50% (30 layers were added, up to a total of 80 layers). These additional numerical simulations include no climate change scenario, and were compared to the current numerical simulations that contained a horizontal resolution of 50 m and 50 layers. The initial conditions of all additional numerical simulations were equal to the calibrated pre-development groundwater heads and TDS concentrations.

The comparison of the current simulations with the three additional numerical simulations that contain higher and lower spatial resolutions (Fig. A1), show a similar increase of the volume of fresh groundwater in the model domain during the simulation period of 2011 to 2050. In the situation with the Sand Engine (Fig. A1b), a coarser spatial resolution lowered the projected volume of fresh groundwater (-10% in 2050), and a finer horizontal and vertical spatial resolution raised the projected volume of groundwater (respectively + 4% and +20% in 2050). However, when taking into account the deviations in the volume of fresh groundwater in the reference case (Fig. A1a), the total change in the volume of fresh groundwater becomes smaller; respectively -2%, +0% and +14% in 2050. The additional simulation with an implicit coupling of flow and transport equations shows a small to negligible difference (smaller than 2% during the entire simulation period) with the simulations with explicit coupling of flow and transport equations (Fig. A2). In summary, the additional simulations show relatively small changes in the volume of fresh groundwater, and suggest that an increase in the vertical resolution can even lead to a higher increase in 
[revised manuscript text omitted]
 The outer model boundaries were situated perpendicular to the coastline (at the SW and NE sides of the modelmodel; Fig. 1), or parallel to the coastline (the NW and SE side of the model). Model boundaries that were defined perpendicular to the coastline, lie parallel to the dominant groundwater flow direction in the coastal area, and were therefore defined as no-flow boundaries. The other Other outer model boundaries (at the NW and SE side of the model) were defined as illustrated in Fig. 4: 'specified-head' and 'headdependent flux' boundary  conditions (taking into account density differences), which represent the  North Sea and local groundwater system, respectively. The 'specified-head' boundary conditions equalled the average level of the North Sea, and were applied to the seafloor. Local groundwater conditions were defined by a previous model simulation of the southwest of the Netherlands (Oude Essink et al., 2010). The base of the model was defined equal to the hydrogeological base of the model domain, which is approximately -170 m MSL and assumed to be a no-flow boundary.

The subsoil of the model was schematised to four aquifers and three aquitards (Fig. 4), based on borehole data, and the national geological databases REGIS II.1 (Vernes and van Doorn, 2005) and GeoTOP (Stafleu et al., 2013) of the Geological Survey of The Netherlands. The upper part of the phreatic aquifer (above -10 MSL) was subdivided into two hydrogeological zones with distinct hydraulic conductivities, because the geological data showed systematic differences in the sediment composition within the model domain: one zone coincides with most of the low-lying polders and contains predominantly clay, loam and fine sand deposits; the other zone contains most of the elevated areas of the model domain, where mainly fine to coarse sand was deposited during the Holocene. The aquifer parameters and layer elevations were defined uniform for each hydrogeological unit, based on parameter estimations in the national hydrogeological database (Table 1). The  molecular diffusion coefficient was set to $10^{-9}$ m$^2$ s$^{-1}$, and the longitudinal dispersivity was set to 0.2 m with a ratio of transversal to longitudinal dispersivity of 0.1. These values are similar to comparable groundwater models in the same region (Eeman et al., 2011; de Louw et al., 2011; Vandenbohede and Lebbe, 2007, 2012).

~~The phreatic aquifer contained two separate hydrogeological units and was subdivided in two hydrogeological layers, because the geological data showed systematic differences in the sediment composition within the model domain. In low lying polders and in the lower section of the phreatic aquifer (between -10 and -16 m MSL), predominantly clay, loam and fine sand was deposited during the Holocene. In the elevated areas of the model domain and the upper section of the phreatic aquifer predominantly fine to coarse sand was deposited. The hydraulic conductivities in areas and layers with mainly fine to coarse sand (beach, dunes and urban area) were set to 10 m d$^{-1}$, and the hydraulic conductivities in areas and layers with mainly clay, loam and fine sand were set to 1 m d$^{-1}$.~~

[revised manuscript text omitted]
 et al., 2015; Mahmoodzadeh et al., 2014). In addition, studies have shown that an increase in the frequency of storm

surge overwash will exacerbate the salinization of coastal aquifers, although the freshwater lens is generally able to recover over time (Holding et al., 2015; Terry and Falkland, 2010). In relation to the morphological dynamics of coastal regions, studies have shown that the erosion and accretion of sand can lead to substantial changes within the beach foredune area (Bakker et al., 2012; Keijsers et al., 2014b), and that climate change might exacerbate coastal erosion (FitzGerald et al., 2008; Zhang et al., 2004). Morphological developments in coastal areas and islands can therefore have a substantial effect on existing and future fresh groundwater resources. Coastal management strategies that compensate, limit, or counteract coastal erosion or seawater intrusion may therefore help to protect or expand fresh groundwater resources.

[revised manuscript text omitted]

**Appendix A: Grid convergence test**

In order to justify that the chosen spatial and temporal discretisation is adequate for reliable numerical quantifications of the potential effect of the Sand Engine on fresh groundwater resources, we have executed a grid convergence test for period 2011 to 2050 (with and without Sand Engine). Current numerical simulations were performed with: a horizontal grid size of 50 m, 50 layers with a variable thickness from 1 (upper layers) to 10 m (lower layers), and stress periods of 90.25 to 92 days (corresponding to seasons). The transport simulations were performed implicitly with the Generalized Conjugate Gradient (GCG) solver, and the transport step sizes were model-calculated based on a Courant number of 1. For the coupling of the flow and transport we have used the explicit approach. In the grid convergence test we have performed one additional with an implicit (iterative with a density criterion of 0.2 kg m$^{-3}$) to test the temporal discretisation. The spatial discretisation was tested with three additional numerical simulations with higher and lower spatial resolutions: one simulation with a horizontal grid size of 25 m, one simulation with a horizontal grid size of 100 m, and one simulation with the same horizontal grid size of 50 m and an increased vertical resolution of the upper layers. In the upper part of the model, up to a depth of -50 m MSL, the layer thicknesses were lowered with 50% (30 layers were added, up to a total of 80 layers). These additional numerical simulations include no climate change scenario, and were compared to the current numerical simulations that contained a horizontal resolution of 50 m and 50 layers. The initial conditions of all additional numerical simulations were equal to the calibrated pre-development groundwater heads and TDS concentrations.

The comparison of the current simulations with the three additional numerical simulations that contain higher and lower spatial resolutions (Fig. A1), show a similar increase of the volume of fresh groundwater in the model domain during the simulation period of 2011 to 2050. In the situation with the Sand Engine (Fig. A1b), a coarser spatial resolution lowered the projected volume of fresh groundwater (-10% in 2050), and a finer horizontal and vertical spatial resolution raised the projected volume of groundwater (respectively + 4% and +20% in 2050). However, when taking into account the deviations in the volume of fresh groundwater in the reference case (Fig. A1a), the total change in the volume of fresh groundwater becomes smaller; respectively -2%, +0% and +14% in 2050. The additional simulation with an implicit coupling of flow and transport equations shows a small to negligible difference (smaller than 2% during the entire simulation period) with the simulations with explicit coupling of flow and transport equations (Fig. A2). In summary, the additional simulations show relatively small changes in the volume of fresh groundwater, and suggest that an increase in the vertical resolution can even lead to a higher increase in 
[revised manuscript text omitted]

---

## Author Response (AR2)

**Reply on review of our manuscript HESS-2016-5 "Fresh groundwater resources in a large sand replenishment"**

Dear Editor,

We would like to thank the reviewers again for their effort reviewing our manuscript and their valuable comments.

In the subsequent pages we have explained point by point how we dealt with their comments. The original reviewer comments are presented in bold italic, and our response is presented in normal text. The marked up manuscript version is added to at the end of this document.

Sincerely,

Sebastian Huizer, on behalf of all authors

**Comments Referee #1**

*The changes made by the authors have improved the ms. Nevertheless, there are still three points that need to be addressed more rigorously as discussed below. The numerical numbers refer to my original comments.*

We would like to thank the Referee for the comments, which are highly appreciated.
* * *
*1. Appendix A. The grid convergence test is now somewhat clearer but some questions remain unanswered and some deficiencies persist. In general, the wording in Appendix A is in some ways complicated. For example, why don't you simply list the spatial and temporal discretization schemes in a simple table, so that the reader has a quick overview of all simulations you ran to attain grid convergence? Also, it must be clearly stated that different time-step sizes were not tested because apparently, the time-step size is always Courant number controlled. So strictly speaking, convergence of the solution with regard to temporal discretization is not shown. L6f: Here, you are using the term implicit and GCG in the same phrase, but the two have nothing to do with each other. "Implicit" denotes a time weighting scheme, and "GCG" is a type of matrix solver. It appears here that the authors are erroneously using numerical terms.*

We have added a statement that time-step sizes were not tested, because the time-step sizes are controlled with stability constraints and accuracy requirements. And to clarify the wording in Appendix A and avoid confusion, we have deleted the reference the GCG solver. We have also added a Table with the additional spatial discretisation schemes.
* * *
*The same applies to L23ff: It appears that "explicit coupling" and "implicit coupling" are ill-used. Physically coupled processes (like flow and transport) can in fact be explicitly coupled, which is called a sequential iterative approach. In this case, coupled equations are solved independently from each other, but they are coupled through equations of state. Physically coupled processes can also be implicitly coupled, which is called a one-step or direct approach. In that case, coupled equations are in fact solved in a single step, requiring a single system matrix where the entries contain contributions of both coupled and disretized equations. I do not believe that this is what the authors mean because different software packages are used and because the underlying equations are probably unknown. I think by "explicit coupling" or "implicit coupling", the authors simply mean that the underlying time-weighting scheme is explicit or implicit. But this is something totally different, and it has nothing to do with the coupling scheme of the equations! The authors should re-think all this, and correctly formulate the numerical details in the ms accordingly.*

The terms "explicit coupling" and "implicit coupling" refer to the coupling of flow and transport equations, and these terms were taken from the description of the model code SEAWAT in:

*Guo, Weixing and Langevin, Christian. D., 2002, User's Guide to SEAWAT: A Computer Program for Simulation of Three-Dimensional Variable-Density Ground-Water Flow: U.S. Geological Survey Techniques of Water-Resources Investigations 6, Chapter A7, 78 p.*

In this technical description of the SEAWAT program, "explicit coupling" refers to a direct coupling of flow and transport equations to equations of state, and "implicit coupling" refers to the solution of the flow and transport equations with an iterative technique (also coupled with equations of state). For more information we refer to the model description.

In Appendix A we have referred to this model description for the description of the explicit and implicit coupling of the flow and transport equations.

*Also, results from Fig. A1 do not convincingly show that grid effects are excluded. A grid convergence test requires a criterion, for which error the result has converged, and thus, which grid is then selected. That criterion is not discussed and an error analysis is absent. Grid convergence is therefore not shown. This must be improved.*

In order to show the possible effects of the discretisation more convincingly we have calculated the Grid Convergence Index (GCI). This incorporates an analysis of the errors.

*5. Section 3.3 somewhat clarifies the deformation story, but not entirely. I understand that this ms will be read mostly by modelers who will understand terms like "re-meshing" or "grid deformation". I think these terms more appropriately describe your numerical approach.*

We have adapted the text to clarify the incorporation of the changes in bathymetry in the numerical simulations, and we agree that the ms will probably by mostly read by modelers and have added the term 'grid regeneration' to more appropriately describe the approach.

*6. Section 4.1 still does not clearly state that the transient model is uncalibrated. For scientific integrity, this must be said.*

We have added a clear statement to section 4.1 that the transient model was (strictly speaking) not calibrated.

[revised manuscript text omitted]

**Legend**
- Sand, clay and peat deposits (Holocene)
- Fluvial and marine sand deposits (Pleistocene)
- Fluvial clay deposits (Early Pleistocene Stamproy Formation)
- Fluvial clay deposits (Early Pleistocene Waalre Formation)
- Marine sand deposits with clay layers (Early Pleistocene Maassluis Formation)

**Fig. 3.** Geological profiles (based on the databases of the Geological Survey of The Netherlands) across the model domain with conceptual fresh-salt water distribution (locations are shown in **Fig. 2**)

[Figure]

**Fig. 4.** Conceptual representation of a slice of the model with layer thicknesses and boundary conditions

[Figure]

**Fig. 5.** Simulation of historical coastal erosion (based on paleogeographic maps of Vos and de Vries, 2013), sea-level rise (black line) and groundwater extraction (blue) in the period 1810 – 2010; dashed lines indicate estimates, and vertical grey lines refer to stress periods (CBS et al., 2013).

[Figure]

**Fig. 6.** Simulated morphological development of the Sand Engine from 2011 to 2050, illustrated by contour maps with the terrain elevation (m MSL)

[Figure]

**Fig. 7.** Comparison of observed and simulated heads (a) and concentration (b)

[Figure]

**Fig. 8.** Phreatic groundwater level (left) and fresh-brackish interface depth (right) after calibration, before the construction of the Sand Engine (2010 – 2011).

[Figure]

**Fig. 9.** Comparison of (a) average groundwater heads in May-June 2014 and (b) (single) TDS concentrations of soil samples taken between 10 and 14 March 2014 with model simulations in the Sand Engine.

[Figure]

**Fig. 10.** Increase of the volume of fresh groundwater in the situation without Sand Engine (a) and situation with Sand Engine (b) in the period 2011 to 2050, where the legend refers to (climate) scenarios (as mentioned in Table 3)

[Figure]

**Fig. 11.** Transects with the simulated groundwater salinity (in g TDS L-1) in 2010 (pre-development Sand Engine), for transect A and B (as shown in Fig. 2 and Fig. 3)

[Figure]

5 **Fig. 12.** Transects with the simulated groundwater salinity (in g TDS L$^{-1}$) in 2050 (including Sand Engine, No Climate Change), for transect A and B (as shown in Fig. 2 and Fig. 3)

[Figure]

**Fig. 13.** Thickness of fresh groundwater [m] in reference scenario near the Sand Engine from 2011 – 2050

[Figure]

**Fig. A1.** Increase of the volume of fresh groundwater in the situation without Sand Engine (a) and situation with Sand Engine (b) in the period 2011 to 2050, where the legend refers to the four grid discretisation simulations

[Figure]

5    **Fig. A2.** Increase of the volume of fresh groundwater in the situation without Sand Engine (a) and situation with Sand Engine (b) in the period 2011 to 2050, where the legend refers to the coupling of flow and transport equations

**Table A1. Grid Convergence Index (GCI) of the simulated increase in the volume of fresh groundwater in 2050 (situation with Sand Engine), for three simulations that contain different spatial grid refinements (S1 to S3).**

| Sim. | Grid size [m] | Layers | Refinement ratio[a] [-] | Relative error[a] [-] | Order of accuracy[a,b] [-] | GCI [%] |
|------|---------------|--------|--------------------------|------------------------|-----------------------------|---------|
| S1 | 25 x 25 | 50 | 1.585 | 0.0224 | 3.36 | 0.76 |
| S2 | 100 x 100 | 50 | 1.585 | 0.1074 | 3.36 | 3.63 |
| S3 | 50 x 50 | 80 | 1.286 | 0.1487 | 3.36[c] | 14.0 |

[a] The refinement ratio, relative error and order or accuracy are parameters that were used to determine the GCI (Roache, 1994). [b] The order of accuracy was estimated with the relative error and refinement ratio of model simulation S1 and S2 (Stern et al., 2001). [c] For model simulation S3 the estimated order of accuracy of simulation S1 and S2 was used to calculate the GCI.